# Learning Group Importance using the Differentiable Hypergeometric Distribution

**Thomas M. Sutter, Laura Manduchi, Alain Ryser, Julia E. Vogt**
Department of Computer Science
ETH Zurich
Switzerland
{thomas.sutter,laura.manduchi,alain.ryser,julia.vogt}@inf.ethz.ch

## Abstract

Partitioning a set of elements into subsets of a priori unknown sizes is essential in many applications. These subset sizes are rarely explicitly learned - be it the cluster sizes in clustering applications or the number of shared versus independent generative latent factors in weakly-supervised learning. Probability distributions over correct combinations of subset sizes are non-differentiable due to hard constraints, which prohibit gradient-based optimization. In this work, we propose the differentiable hypergeometric distribution. The hypergeometric distribution models the probability of different group sizes based on their relative importance. We introduce reparameterizable gradients to learn the importance between groups and highlight the advantage of explicitly learning the size of subsets in two typical applications: weakly-supervised learning and clustering. In both applications, we outperform previous approaches, which rely on suboptimal heuristics to model the unknown size of groups.

## 1 Introduction

Many machine learning approaches rely on differentiable sampling procedures, from which the reparameterization trick for Gaussian distributions is best known (Kingma & Welling, 2014; Rezende et al., 2014). The non-differentiable nature of discrete distributions has long hindered their use in machine learning pipelines with end-to-end gradient-based optimization. Only the concrete distribution (Maddison et al., 2017) or Gumbel-Softmax trick (Jang et al., 2016) boosted the use of categorical distributions in stochastic networks. Unlike the high-variance gradients of score-based methods such as REINFORCE (Williams, 1992), these works enable reparameterized and low-variance gradients with respect to the categorical weights. Despite enormous progress in recent years, the extension to more complex probability distributions is still missing or comes with a trade-off regarding differentiability or computational speed (Huijben et al., 2021).

The hypergeometric distribution plays a vital role in various areas of science, such as social and computer science and biology. The range of applications goes from modeling gene mutations and recommender systems to analyzing social networks (Becchetti et al., 2011; Casiraghi et al., 2016; Lodato et al., 2015). The hypergeometric distribution describes sampling without replacement and, therefore, models the number of samples per group given a limited number of total samples. Hence, it is essential wherever the choice of a single group element influences the probability of the remaining elements being drawn. Previous work mainly uses the hypergeometric distribution implicitly to model assumptions or as a tool to prove theorems. However, its hard constraints prohibited integrating the hypergeometric distribution into gradient-based optimization processes.

In this work, we propose the differentiable hypergeometric distribution. It enables the reparameterization trick for the hypergeometric distribution and allows its integration into stochastic networks of modern, gradient-based learning frameworks. In turn, we learn the size of groups by modeling their relative importance in an end-to-end fashion. First, we evaluate our approach by performing a Kolmogorov-Smirnov test, where we compare the proposed method to a non-differentiable reference implementation. Further, we highlight the advantages of our new formulation in two different applications, where previous work failed to learn the size of subgroups of samples explicitly. Our

first application is a weakly-supervised learning task where two images share an unknown number of generative factors. The differentiable hypergeometric distribution learns the number of shared and independent generative factors between paired views through gradient-based optimization. In contrast, previous work has to infer these numbers based on heuristics or rely on prior knowledge about the connection between images. Our second application integrates the hypergeometric distribution into a variational clustering algorithm. We model the number of samples per cluster using an adaptive hypergeometric distribution prior. By doing so, we overcome the simplified *i.i.d.* assumption and establish a dependency structure between dataset samples. The contributions of our work are the following: i) we introduce the differentiable hypergeometric distribution, which enables its use for gradient-based optimization, ii) we demonstrate the accuracy of our approach by evaluating it against a reference implementation, and iii) we show the advantages of explicitly learning the size of groups in two different applications, namely weakly-supervised learning and clustering.

## 2   RELATED WORK

In recent years, finding continuous relaxations for discrete distributions and non-differentiable algorithms to integrate them into differentiable pipelines gained popularity. Jang et al. (2016) and Maddison et al. (2017) concurrently propose the Gumbel-Softmax gradient estimator. It enables reparameterized gradients with respect to parameters of the categorical distribution and their use in differentiable models. Methods to select $k$ elements - instead of only one - are subsequently introduced. Kool et al. (2019; 2020a) implement sequential sampling without replacement using a stochastic beam search. Kool et al. (2020b) extend the sequential sampling procedure to a reparameterizable estimator using REINFORCE. Grover et al. (2019) propose a relaxed version of a sorting procedure, which simultaneously serves as a differentiable and reparameterizable top-$k$ element selection procedure. Xie & Ermon (2019) propose a relaxed subset selection algorithm to select a given number $k$ out of $n$ elements. Paulus et al. (2020) generalize stochastic softmax tricks to combinatorial spaces.[1] Unlike Kool et al. (2020b), who also use a sequence of categorical distributions, the proposed method describes a differentiable reparameterization for the more complex but well-defined hypergeometric distribution. Differentiable reparameterizations of complex distributions with learnable parameters enable new applications, as shown in Section 5.

The classical use case for the hypergeometric probability distribution is sampling without replacement, for which urn models serve as the standard example. The hypergeometric distribution has previously been used as a modeling distribution in simulations of social evolution (Ono et al., 2003; Paolucci et al., 2006; Lashin et al., 2007), tracking of human neurons and gene mutations (Lodato et al., 2015; 2018), network analysis (Casiraghi et al., 2016), and recommender systems (Becchetti et al., 2011). Further, it is used as a modeling assumption in submodular maximization (Feldman et al., 2017; Harshaw et al., 2019), multimodal VAEs (Sutter & Vogt, 2021), k-means clustering variants (Chien et al., 2018), or random permutation graphs (Bhattacharya & Mukherjee, 2017). Despite not being differentiable, current sampling schemes for the multivariate hypergeometric distribution are a trade-off between numerical stability and computational efficiency (Liao & Rosen, 2001; Fog, 2008a;b).

## 3   PRELIMINARIES

Suppose we have an urn with marbles in different colors. Let $c \in \mathbb{N}$ be the number of different classes or groups (e.g. marble colors in the urn), $\boldsymbol{m} = [m_1, \ldots, m_c] \in \mathbb{N}^c$ describe the number of elements per class (e.g. marbles per color), $N = \sum_{i=1}^{c} m_i$ be the total number of elements (e.g. all marbles in the urn) and $n \in \{0, \ldots, N\}$ be the number of elements (e.g. marbles) to draw. Then, the multivariate hypergeometric distribution describes the probability of drawing $\boldsymbol{x} = [x_1, \ldots, x_c] \in \mathbb{N}_0^c$ marbles by sampling without replacement such that $\sum_{i=1}^{c} x_i = n$, where $x_i$ is the number of drawn marbles of class $i$.

Using the *central* hypergeometric distribution, every marble is picked with equal probability. The number of selected elements per class is then proportional to the ratio between number of elements per class and the total number of elements in the urn. This assumption is often too restrictive, and we

---

[1]Huijben et al. (2021) provide a great review article of the Gumbel-Max trick and its extensions describing recent algorithmic developments and applications.

want an additional modeling parameter for the importance of a class. Generalizations, which make certain classes more likely to be picked, are called *noncentral* hypergeometric distributions.

In the literature, two different versions of the noncentral hypergeometric distribution exist, Fisher's (Fisher, 1935) and Wallenius' (Wallenius, 1963; Chesson, 1976) distribution. Due to limitations of the latter (Fog, 2008a), we will refer to Fisher's version of the noncentral hypergeometric distribution in the remaining part of this work.

**Definition 3.1** (Multivariate Fisher's Noncentral Hypergeometric Distribution (Fisher, 1935))**.** A random vector $\boldsymbol{X}$ follows Fisher's noncentral multivariate distribution, if its joint probability mass function is given by

$$P(\boldsymbol{X} = \boldsymbol{x}; \boldsymbol{\omega}) = p_{\boldsymbol{X}}(\boldsymbol{x}; \boldsymbol{\omega}) = \frac{1}{P_0} \prod_{i=1}^{c} \binom{m_i}{x_i} \omega_i^{x_i} \tag{1}$$

where $P_0 = \sum_{\boldsymbol{y} \in \mathcal{S}} \prod_{i=1}^{c} \binom{m_i}{y_i} \omega_i^{y_i}$. The support $\mathcal{S}$ of the PMF is given by $\mathcal{S} = \{\boldsymbol{x} \in \mathbb{N}_0^c : \forall i \quad x_i \leq m_i, \sum_{i=1}^{c} x_i = n\}$ and $\binom{n}{k} = \frac{n!}{k!(n-k)!}$.

The total number of samples per class $\boldsymbol{m}$, the number of samples to draw $n$, and the class importance $\boldsymbol{\omega}$ parameterize the multivariate distribution. Here, we assume $\boldsymbol{m}$ and $n$ to be constant per experiment and are mainly interested in the class importance $\boldsymbol{\omega}$. Consequently, we only use $\boldsymbol{\omega}$ as the distribution parameter in Equation (1) and the remaining part of this work.

The class importance $\boldsymbol{\omega}$ is a crucial modeling parameter in applying the noncentral hypergeometric distribution (see (Chesson, 1976)). It resembles latent factors like the importance, fitness, or adaptation capabilities of a class of elements, which are often more challenging to measure in field experiments than the sizes of different populations. Introducing a differentiable and reparameterizable formulation enables the learning of class importance from data (see Section 5). We provide a more detailed introduction to the hypergeometric distribution in Appendix A.

## 4 METHOD

The reparameterizable sampling for the proposed differentiable hypergeometric distribution consists of three parts:

1. Reformulate the multivariate distribution as a sequence of interdependent and conditional univariate hypergeometric distributions.
2. Calculate the probability mass function of the respective univariate distributions.
3. Sample from the conditional distributions utilizing the Gumbel-Softmax trick.

We explain all steps in the following Sections 4.1 to 4.3. Additionally, Algorithm 1 and Algorithm 2 (see Appendix B.5) describe the full reparameterizable sampling method using pseudo-code and Figures 5 and 6 in the appendix illustrate it graphically.

### 4.1 SEQUENTIAL SAMPLING USING CONDITIONAL DISTRIBUTIONS

Because it scales linearly with the number of classes and not with the size of the support $\mathcal{S}$ (see Definition 3.1), we use the conditional sampling algorithm (Liao & Rosen, 2001; Fog, 2008b). Following the chain rule of probability, we sample from the following sequence of conditional probability distributions

$$p_{\boldsymbol{X}}(\boldsymbol{x}; \boldsymbol{\omega}) = p_{X_1}(x_1; \boldsymbol{\omega}) \prod_{i=2}^{c} p_{X_i}(x_i | \{\bigcup_{j<i} x_j\}; \boldsymbol{\omega}) \tag{2}$$

Following Equation (2), every $p_{X_i}(\cdot)$ describes the probability of a single class $i$ of samples given the already sampled classes $j < i$. In the conditional sampling method, we model every conditional distribution $p_{X_i}(\cdot)$ as a univariate hypergeometric distribution with two classes $L$ and $R$: for $i = 1..c$, we define class $L := \{i\}$ as the left class and class $R := \{j : j > i \land j \leq c\}$ as the right class.

**Algorithm 1** Sampling from the differentiable hypergeometric distribution. The different blocks are explained in more detail in Sections 4.1 to 4.3 and Algorithm 2.

---

**Input:** $\boldsymbol{m} \in \mathbb{N}^c, \boldsymbol{\omega} \in \mathbb{R}^c_{0+}, n \in \mathbb{N}, \tau \in \mathbb{R}_{0+}$
**Output:** $\boldsymbol{x} \in \mathbb{N}^c_0, \{\boldsymbol{\alpha}_i \in \mathbb{R}^{m_i}\}^c_{i=1}, \{\hat{\boldsymbol{r}}_i \in \mathbb{R}^{m_i}\}^c_{i=1}$
**for** $i \in \{1, \dots, c\}$ **do**
    $L \leftarrow i, R \leftarrow \{\bigcup^c_{j=i+1} j\}$               # Formulate the multivariate as a univariate
    $\boldsymbol{m} \rightarrow m_L, m_R \in \mathbb{Z}_{0+}, \boldsymbol{\omega} \rightarrow \omega_L, \omega_R \in \mathbb{R}_{0+}$           # distribution (Section 4.1)
    $x_L, \boldsymbol{\alpha}_L, \hat{\boldsymbol{r}}_L \leftarrow$ sampleUNCHG$(m_L, m_R, \omega_L, \omega_R, n, \tau)$   # Sample from univariate distribution
    $n \leftarrow n - x_L, \boldsymbol{m} \leftarrow \boldsymbol{m} \setminus \boldsymbol{m}_L, \boldsymbol{\omega} \leftarrow \boldsymbol{\omega} \setminus \omega_L$          # Re-assign classes for next step
    $x_i \leftarrow x_L, \boldsymbol{\alpha}_i \leftarrow \boldsymbol{\alpha}_L, \hat{\boldsymbol{r}}_i \leftarrow \hat{\boldsymbol{r}}_L$                 # Assign values for class $i$
**end for**
**return** $\boldsymbol{x}, \{\boldsymbol{\alpha}_i\}^c_{i=1}, \{\hat{\boldsymbol{r}}_i\}^c_{i=1}$

**function** SAMPLEUNCHG$(m_i, m_j, \omega_i, \omega_j, n, \tau)$
    $\boldsymbol{\alpha}_i \leftarrow$ calcLogPMF$(m_i, m_j, \omega_i, \omega_j, n)$                           # Section 4.2
    $x_i, \hat{\boldsymbol{r}}_i \leftarrow$ contRelaxSample$(\boldsymbol{\alpha}_i, \tau))$                           # Section 4.3
    **return** $x_i, \boldsymbol{\alpha}_i, \hat{\boldsymbol{r}}_i$
**end function**

---

To sample from the original multivariate hypergeometric distribution, we sequentially sample from the urn with only two classes $L$ and $R$, which simplifies to sampling from the univariate noncentral hypergeometric distribution given by the following parameters (Fog, 2008b):

$$m_L = \sum_{l \in L} m_l, \qquad m_R = \sum_{r \in R} m_r, \qquad \omega_L = \frac{\sum_{l \in L} \omega_l \cdot m_l}{m_L}, \qquad \omega_R = \frac{\sum_{r \in R} \omega_r \cdot m_r}{m_R} \qquad (3)$$

We leave the exploration of different and more sophisticated subset selection strategies for future work.

Samples drawn using this algorithm are only approximately equal to samples from the joint noncentral multivariate distribution with equal $\tilde{\boldsymbol{\omega}}$. Because of the merging operation in Equation (3), the approximation error is only equal to zero for the central hypergeometric distribution. One way to reduce this approximation error independent of the learned $\boldsymbol{\omega}$ is a different subset selection algorithm (Fog, 2008b). Note that the proposed method introduces an approximation error compared to a non-differentiable reference implementation with the same $\boldsymbol{\omega}$ (see Section 5.1), but not the underlying and desired true class importance. We can still recover the true class importance because a different $\boldsymbol{\omega}$ overcomes the approximation error introduced by merging needed for the conditional sampling procedure.

## 4.2 CALCULATE PROBABILITY MASS FUNCTION

In Section 4.1, we derive a sequential sampling procedure for the hypergeometric distribution, in which we repeatedly draw from a univariate distribution to simplify sampling. Therefore, we only need to compute the PMF for the univariate distribution. See Appendix B.1, for the multivariate extension. The PMF $p_{X_L}(x_L; \boldsymbol{\omega})$ for the hypergeometric distribution of two classes $L$ and $R$ defined by $m_L, m_R, \omega_L, \omega_R$ and $n$ is given as

$$p_{X_L}(x_L; \boldsymbol{\omega}) = \frac{1}{P_0} \binom{m_L}{x_L} \omega_L^{x_L} \binom{m_R}{n - x_L} \omega_R^{n - x_L} \qquad (4)$$

$P_0$ is defined as in Equation (1), $\omega_L, \omega_R$ and their derivation from $\boldsymbol{\omega}$, and $m_L, m_R, n$ as in Equation (3). The large exponent of $\boldsymbol{\omega}$ and the combinatorial terms can lead to numerical instabilities making the direct calculation of the PMF in Equation (4) infeasible. Calculations in $\log$-domain increase numerical stability for such large domains, while keeping the relative ordering.

**Lemma 4.1.** *The unnormalized log-probabilities*

$$\log p_{X_L}(x_L; \boldsymbol{\omega}) = x_L \log \omega_L + (n - x_L) \log \omega_R + \psi_F(x_L) + C \qquad (5)$$

*define the unnormalized weights of a categorical distribution that follows Fisher's noncentral hypergeometric distribution. $C$ is a constant and $\psi_F(x_L)$ is defined as*

$$\psi_F(x_L) = -\log\left(\Gamma(x_L + 1)\Gamma(n - x_L + 1)\right) - \log\left(\Gamma(m_L - x_L + 1)\Gamma(m_R - n + x_L + 1)\right) \quad (6)$$

We provide the proof to Lemma 4.1 in Appendix B.2. Common automatic differentiation frameworks[2] have numerically stable implementations of $\log \Gamma(\cdot)$. Therefore, Equation (6) and more importantly Equation (5) can be calculated efficiently and reliably. Lemma 4.1 relates to the `calcLogPMF` function in Algorithm 1, and Algorithm 2 describes `calcLogPMF` in more detail.

Using the multivariate form of Lemma 4.1 (see Appendix B.1), it is possible to directly calculate the categorical weights for the full support $\mathcal{S}$. Compared to the conditional sampling procedure, this would result in a computational speed-up for a large number of classes $c$. However, the size of the support $\mathcal{S}$ is $\prod_{i=1}^{c} m_i$, quickly resulting in unfeasible memory requirements. Therefore, we would restrict ourselves to settings with no practical relevance.

### 4.3 CONTINUOUS RELAXATION FOR THE CONDITIONAL DISTRIBUTION

Continuous relaxations describe procedures to make discrete distributions differentiable with respect to their parameters (Huijben et al., 2021). Following Lemma 4.1, we make use of the Gumbel-Softmax trick to reparameterize the hypergeometric distribution via its conditional distributions $p_{X_L}(\cdot)$. The Gumbel-Softmax trick enables a reparameterization of categorical distributions that allows the computation of gradients with respect to the distribution parameters. We state Lemma 4.2, and provide a proof in Appendix B.3.

**Lemma 4.2.** *The Gumbel-Softmax trick can be applied to the conditional distribution $p_{X_i}(x_i | \{x_k\}_{k=1}^{i-1}; \boldsymbol{\omega})$ of class $i$ given the already sampled classes $k < i$.*

Lemma 4.2 connects the Gumbel-Softmax trick to the hypergeometric distribution. Hence, reparameterizing enables gradients with respect to the parameter $\boldsymbol{\omega}$ of the hypergeometric distribution:

$$\boldsymbol{u} \sim \boldsymbol{U}(0,1), \qquad \boldsymbol{g}_i = -\log(-\log(\boldsymbol{u})), \qquad \hat{\boldsymbol{r}}_i = \boldsymbol{\alpha}_i(\boldsymbol{\omega}) + \boldsymbol{g}_i \qquad (7)$$

where $\boldsymbol{u} \in [0,1]^{m_i+1}$ is a random sample from an *i.i.d.* uniform distribution $\boldsymbol{U}$. $\boldsymbol{g}_i$ is therefore *i.i.d.* gumbel noise. $\hat{\boldsymbol{r}}_i$ are the perturbed conditional probabilities for class $i$ given the class conditional unnormalized log-weights $\boldsymbol{\alpha}_i(\boldsymbol{\omega})$:

$$\boldsymbol{\alpha}_i(\boldsymbol{\omega}) = \log \boldsymbol{p}_{X_i}(\boldsymbol{x}_i; \boldsymbol{\omega}) - C = [\log p_{X_i}(0; \boldsymbol{\omega}), \ldots, \log p_{X_i}(m_i; \boldsymbol{\omega})] - C \qquad (8)$$

We use the softmax function to generate $(m_i + 1)$-dimensional sample vectors from the perturbed unnormalized weights $\hat{\boldsymbol{r}}_i/\tau$, where $\tau$ is the temperature parameter. Due to Lemma 4.2, we do not need to calculate the constant $C$ in Equations (5) and (8). We refer to the original works (Jang et al., 2016; Maddison et al., 2017) or Appendix A.2 for more details on the Gumbel-Softmax trick itself. Lemma 4.2 corresponds to the `contRelaxSample` function in Algorithm 1 (see Algorithm 2 for more details).

Note the difference between the categorical and the hypergeometric distribution concerning the Gumbel-Softmax trick. Whereas the (unnormalized) category weights are also the distribution parameters for the former, the log-weights $\boldsymbol{\alpha}_i$ of class $i$ are a function of the class importance $\boldsymbol{\omega}$ and the pre-defined $\boldsymbol{x}_i = [0, \ldots, m_i]$ for the latter. It follows that for a sequence of categorical distributions, we would have $\sum_{i=1}^{c} m_i$ learnable parameters, whereas for the differentiable hypergeometric distribution we only have $c$ learnable parameters. In Appendix B.4, we discuss further difficulties in using a sequence of unconstrained differentiable categorical distributions.

## 5 EXPERIMENTS

We perform three experiments that empirically validate the proposed method and highlight the versatility and applicability of the differentiable hypergeometric distribution to different important areas of machine learning. We first test the generated samples of the proposed differentiable formulation procedure against a non-differentiable reference implementation. Second, we present how the hypergeometric distribution helps detecting shared generative factors of paired samples in a weakly-supervised setting. Our third experiment demonstrates the hypergeometric distribution as a prior in variational clustering algorithms. In our experiments we focus on applications in the field of distribution inference where we make use of the reparameterizable gradients. Nevertheless, the

---

[2] E. g. Tensorflow (Abadi et al., 2016) or PyTorch (Paszke et al., 2019)

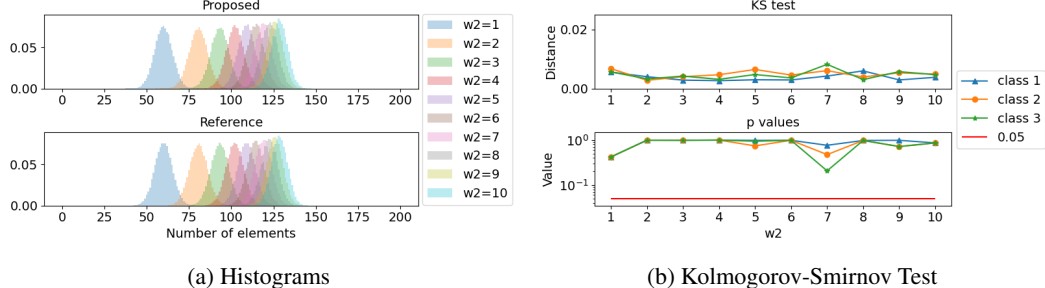

(a) Histograms                                    (b) Kolmogorov-Smirnov Test

Figure 1: Comparing random variables from the proposed differentiable formulation to a non-differentiable reference implementation. We draw samples from a multivariate noncentral hypergeometric distribution consisting of three classes. $m_i = 200 \ \forall i$ and $n = 180$. The class weights $\omega_1$ and $\omega_3$ for classes 1 and 3 are set to $1.0$, $\omega_2$ is increased from $1.0$ to $10.0$ with a step size of $1.0$ (w2 in the figure). Figure 1a shows histograms of the number of elements for class 2. Figure 1b represents the calculated distance values of the KS test between the reference and proposed implementation (upper plot) and their respective $p$-values (lower plot).

proposed method is applicable to any application with gradient-based optimization, in which the underlying process models sampling without replacement [3].

## 5.1 KOLMOGOROV-SMIRNOV TEST

To assess the accuracy of the proposed method, we evaluate it against a reference implementation using the Kolmogorov-Smirnov test (Kolmogorov, 1933; Smirnov, 1939, KS). It is a nonparametric test to estimate the equality of two distributions by quantifying the distance between the empirical distributions of their samples. The null distribution of this test is calculated under the null hypothesis that the two groups of samples are drawn from the same distribution. If the test fails to reject the null hypothesis, the same distribution generated the two groups of samples, i.e., the two underlying distributions are equal. As described in Section 4, we use class conditional distributions to sample from the differentiable hypergeometric distribution. We compare samples from our differentiable formulation to samples from a non-differentiable reference implementation (Virtanen et al., 2020, SciPy). For this experiment, we use a multivariate hypergeometric distribution of three classes. We perform a sensitivity analysis with respect to the class weights $\boldsymbol{\omega}$. We keep $\omega_1$ and $\omega_3$ fixed at $1.0$, and $\omega_2$ is increased from $1.0$ to $10.0$ in steps of $1.0$. For every value of $\omega_2$, we sample $50000$ *i.i.d.* random vectors. We use the Benjamini-Hochberg correction (Benjamini & Hochberg, 1995) to adjust the $p$-values for false discovery rate of multiple comparisons as we are performing $c = 3$ tests per joint distribution. Given a significance threshold of $t = 0.05$, $p > 0.05$ implies that we cannot reject the null hypothesis, which is desirable for our application.

Figure 1a shows the histogram of class 2 samples for all values of $\omega_2$, and Figure 1b the results of the KS test for all classes. The histograms for classes 1 and 3 are in the Appendix (Figure 4). We see that the calculated distances of the KS-test are small, and the corrected $p$-values well above the threshold. Many are even close to $1.0$. Hence, the test clearly fails to reject the null hypothesis in 30 out of 30 cases. Additionally, the proposed and the reference implementation histograms are visually similar. The results of the KS test strongly imply that the proposed differentiable formulation effectively follows a noncentral hypergeometric distribution. We provide more analyses and results from KS test experiments in Appendix C.1.

## 5.2 WEAKLY-SUPERVISED LEARNING

Many data modalities, such as consecutive frames in a video, are not observed as *i.i.d.* samples, which provides a weak-supervision signal for representation learning and generative models. Hence, we are not only interested in learning meaningful representations and approximating the data distribution but also in the detailed relation between frames. Assuming underlying factors generate such coupled

---

[3]The code can be found here: `https://github.com/thomassutter/mvhg`

Table 1: To compare the three methods (LabelVAE, AdaVAE, HGVAE) in the weakly-supervised experiment, we evaluate their learned latent representations with respect to shared (S) and independent (I) generative factors. To assess the amount of shared and independent information in the latent representation, we train linear classifiers on the respective latent dimensions only. We report the adjusted balanced classification accuracy, such that the random classifier achieves score 0.

| | $s = 0$ | $s = 1$ | | $s = 3$ | | $s = 5$ | |
|---|---|---|---|---|---|---|---|
| | I | S | I | S | I | S | I |
| LABEL | $0.14_{\pm 0.01}$ | $0.19_{\pm 0.03}$ | $0.16_{\pm 0.01}$ | $\mathbf{0.10}_{\pm 0.00}$ | $0.23_{\pm 0.01}$ | $\mathbf{0.34}_{\pm 0.00}$ | $0.00_{\pm 0.00}$ |
| ADA | $0.12_{\pm 0.01}$ | $0.19_{\pm 0.01}$ | $0.15_{\pm 0.01}$ | $\mathbf{0.10}_{\pm 0.03}$ | $0.22_{\pm 0.02}$ | $0.33_{\pm 0.03}$ | $0.00_{\pm 0.00}$ |
| HG (OURS) | $\mathbf{0.18}_{\pm 0.01}$ | $\mathbf{0.22}_{\pm 0.05}$ | $\mathbf{0.19}_{\pm 0.01}$ | $0.08_{\pm 0.02}$ | $\mathbf{0.28}_{\pm 0.01}$ | $0.28_{\pm 0.01}$ | $\mathbf{0.01}_{\pm 0.00}$ |

data, a subset of factors should be shared among frames to describe the underlying concept leading to the coupling. Consequently, differing concepts between coupled frames stem from independent factors exclusive to each frame. The differentiable hypergeometric distribution provides a principled approach to learning the number of shared and independent factors in an end-to-end fashion.

In this experiment, we look at pairs of images from the synthetic *mpi3D* toy dataset (Gondal et al., 2019). We generate a coupled dataset by pairing images, which share a certain number of their seven generative factors. We train all models as variational autoencoders (Kingma & Welling, 2014, VAE) to maximize an evidence lower bound (ELBO) on the marginal log-likelihood of images using the setting and code from Locatello et al. (2020). We compare three methods, which sequentially encode both images to some latent representations using a single encoder. Based on the two representations, the subset of shared latent dimensions is aggregated into a single representation. Finally, a single decoder independently computes reconstructions for both samples based on the imputed shared and the remaining independent latent factors provided by the respective encoder. The methods only differ in how they infer the subset of shared latent factors. We refer to Appendix C.2 or Locatello et al. (2020) for more details on the setup of the weakly-supervised experiment.

The LabelVAE assumes that the number of independent factors is known (Bouchacourt et al., 2018; Hosoya, 2018, LabelVAE). Like in Locatello et al. (2020), we assume 1 known factor for all experiments. The AdaVAE relies on a heuristic to infer the shared factors (Locatello et al., 2020, AdaVAE). Based on the Kullback-Leibler (KL) divergence between corresponding latent factors across images, the decision between shared and independent factors is based on a hand-designed threshold. The proposed HGVAE uses the differentiable hypergeometric distribution to model the number of shared and independent latent factors. We infer the unknown group importance $\boldsymbol{\omega} \in \mathbb{R}_{0+}^2$ with a single dense layer, which uses the KL divergences between corresponding latent factors as input (similar to AdaVAE). Based on $\boldsymbol{\omega}$ and with $d$ being the number of latent dimensions, we sample random variables to estimate the $k$ independent and $s := d - k$ shared factors by reparameter-

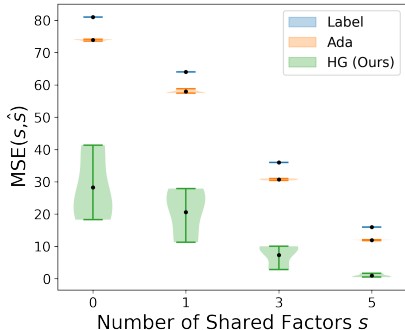

Figure 2: We report the mean squared error MSE$(s, \hat{s})$ between true $s$ and estimated $\hat{s}$ number of shared factors to assess the models' performance.

izing the hypergeometric distribution, providing us with $\hat{k}$ and $\hat{s} := d - \hat{k}$. The proposed differentiable formulation allows us to infer such $\boldsymbol{\omega}$ and simultaneously learn the latent representation in a fully differentiable setting. After sorting the latent factors by KL divergence (Grover et al., 2019), we define the top-$\hat{k}$ latent factors as independent, and the remaining $\hat{s}$ as shared ones. For a more detailed description of HGVAE, the baseline models, and the dataset, see Appendix C.2.

To evaluate the methods, we compare their performance on two different tasks. We measure the mean squared error MSE$(s, \hat{s})$ between the actual number of shared latent factors $s$ and the estimated number $\hat{s}$ (Figure 2) and the classification accuracy of predicting the generative factors on the shared and independent subsets of the learned representations (Table 1). We train classifiers for all factors on the shared and independent part of the latent representation and calculate their balanced accuracy. The reported scores are the average over the factor-specific balanced accuracies. Because the number of different classes differs between the discrete generative factors, we report the adjusted balanced accuracy as a classification metric. These two tasks challenge the methods regarding their estimate

of the relationship between images. Generating the dataset controls the number of independent and shared factors, $k$ and $s$, which allows us to evaluate the methods on different versions of the same underlying data regarding the number of shared and independent generative factors. We generate four weakly-supervised datasets with $s = \{0, 1, 3, 5\}$. On purpose, we also evaluate the edge case of $s = 0$, which is equal to the two views not sharing any generative factors.

Figure 2 shows that previous methods cannot accurately estimate the number of shared factors. Both baseline methods estimate the same number of shared factors $\hat{s}$ independent of the underlying ground truth number of shared factors $s$. What is not surprising for the first model is unexpected for the second approach, given their - in theory - adaptive heuristic. On the other hand, the low mean squared error (MSE) reflects that the proposed HGVAE can dynamically estimate the number of shared factors for every number of shared factors $s$. These results suggest that the differentiable hypergeometric distribution is able to learn the relative importance of shared and independent factors in the latent representations. Previous works' assumptions, though, do not reflect the data's generative process (LabelVAE), or the designed heuristics are oversimplified (AdaVAE). We also see the effect of these oversimplified assumptions in evaluating the learned latent representation. Table 1 shows that the estimation of a large number of shared factors $\hat{s}$ leads to an inferior latent representation of the independent factors, which is reflected in the lower accuracy scores of previous work compared to the proposed method. More surprisingly, for the shared latent representation, our HGVAE reaches the same performance on the shared latent representation despite being more flexible.

Given the general nature of the method, the positive results of the proposed method are encouraging. Unlike previous works, it is not explicitly designed for weakly-supervised learning but achieves results that are more than comparable to field-specific models. Additionally, the proposed method accurately estimates the latent space structure for different experimental settings.

## 5.3 DEEP VARIATIONAL CLUSTERING

We investigate the use of the differentiable hypergeometric distribution in a deep clustering task. Several techniques have been proposed in the literature to combine long-established clustering algorithms, such as K-means or Gaussian Mixture Models, with the flexibility of deep neural networks to learn better representations of high-dimensional data (Min et al., 2018). Among those, Jiang et al. (2016), Dilokthanakul et al. (2016), and Manduchi et al. (2021) include a trainable Gaussian Mixture prior distribution in the latent space of a VAE. A Gaussian Mixture model permits a probabilistic approach to clustering where an explicit generative assumption of the data is defined.

All methods are optimized using stochastic gradient variational Bayes (Kingma & Welling, 2014; Rezende et al., 2014). A major drawback of the above models is that the samples are either assumed to be *i.i.d.* or they require pairwise side information, which limits their applicability in real-world scenarios.

The differentiable hypergeometric distribution can be easily integrated in VAE-based clustering algorithms to overcome limitations of current approaches. We want to cluster a given dataset $X = \{\mathbf{x}_i\}_{i=1}^N$ into $K$ subgroups. Like previous work (Jiang et al., 2016), we assume the data is generated from a random process where the cluster assignments are first drawn from a prior

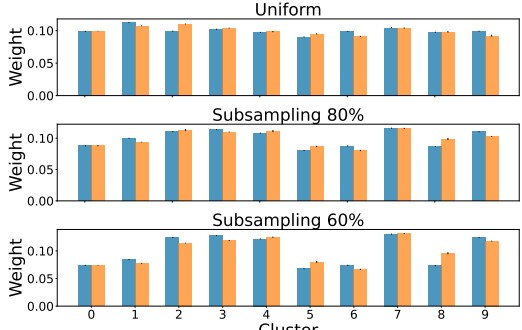

Figure 3: True class (■) vs. learned hypergeometric cluster weights $\pi_i$ (■)

probability $p(\mathbf{c}; \boldsymbol{\pi})$, then each latent embedding $\mathbf{z}_i$ is sampled from a Gaussian distribution, whose mean and variance depend on the selected cluster $c_i$. Finally the sample $\mathbf{x}_i$ is generated from a Bernoulli distribution whose parameter $\boldsymbol{\mu}_{x_i}$ is the output of a neural network parameterized by $\boldsymbol{\theta}$, as in the classical VAE. With these assumptions, the latent embeddings $\mathbf{z}_i$ follow a mixture of Gaussian distributions, whose means and variances, $\{\boldsymbol{\mu}_i, \boldsymbol{\sigma}_i^2\}_{i=1}^K$, are tunable parameters. The above generative model can then be optimised by maximising the ELBO using the stochastic gradient variational Bayes estimator (we refer to Appendix C.3 for a complete description of the optimisation procedure). Previous work (Jiang et al., 2016) modeled the prior distribution as $p(\mathbf{c}; \boldsymbol{\pi}) = \prod_i p(c_i) = \prod_i Cat(c_i \mid \boldsymbol{\pi})$ with either tunable or fixed parameters $\boldsymbol{\pi}$. In this task, we instead replace this prior

Table 2: Evaluation of the clustering experiment on the MNIST datasets. We compare the methods on 3 different dataset versions, namely i) uniform class distribution ii) subsampling with 80% of samples and iii) subsampling with only 60% of samples. We subsample half of the classes. Accuracy (Acc), normalized mutual information (NMI), and adjusted rand index (ARI) are used as evaluation metrics. Higher is better for all metrics. Mean and standard deviations are computed across 5 runs. For fair comparison with the baselines all methods use the pretraining weights provided by Jiang et al. (2016).

| DATASET VERSION | | UNIFORM | CATEGORICAL | HYPERGEOMETRIC |
|---|---|---|---|---|
| UNIFORM | ACC (%) | $\mathbf{92.0 \pm 3.0}$ | $87.2 \pm 5.0$ | $91.4 \pm 5.0$ |
| | NMI (%) | $84.8 \pm 2.2$ | $81.8 \pm 1.9$ | $\mathbf{85.6 \pm 2.0}$ |
| | ARI (%) | $84.2 \pm 4.3$ | $78.3 \pm 4.6$ | $\mathbf{84.8 \pm 4.6}$ |
| SUBSAMPLING 80% | ACC (%) | $90.8 \pm 4.0$ | $87.4 \pm 4.7$ | $\mathbf{92.5 \pm 0.5}$ |
| | NMI (%) | $84.1 \pm 2.2$ | $81.8 \pm 2.3$ | $\mathbf{84.6 \pm 0.8}$ |
| | ARI (%) | $83.2 \pm 3.6$ | $78.2 \pm 5.0$ | $\mathbf{84.4 \pm 1.0}$ |
| SUBSAMPLING 60% | ACC (%) | $83.5 \pm 3.9$ | $86.5 \pm 4.9$ | $\mathbf{89.7 \pm 4.3}$ |
| | NMI (%) | $80.7 \pm 1.4$ | $81.3 \pm 2.9$ | $\mathbf{82.9 \pm 2.2}$ |
| | ARI (%) | $77.6 \pm 2.6$ | $77.7 \pm 6.3$ | $\mathbf{81.5 \pm 3.9}$ |

with the multivariate noncentral hypergeometric distribution with weights $\boldsymbol{\pi}$ and $K$ classes where every class relates to a cluster. Hence, we sample the number of elements per cluster (or cluster size) following Definition 3.1 and Algorithm 1. The hypergeometric distribution permits to create a dependence between samples. The prior probability of a sample to be assigned to a given cluster is not independent of the remaining samples anymore, allowing us to loosen the over-restrictive *i.i.d.* assumption. We explore the effect of three different prior probabilities in Equation (54), namely (i) the categorical distribution, by setting $p(\mathbf{c}; \boldsymbol{\pi}) = \prod_i Cat(c_i \mid \boldsymbol{\pi})$; (ii) the uniform distribution, by fixing $\pi_i = 1/K \; \forall \; i \in \{1, \ldots, K\}$; and (iii) the multivariate noncentral hypergeometric distribution. We compare them on three different MNIST versions (LeCun & Cortes, 2010). The first version is the standard dataset which has a balanced class distribution. For the second and third dataset version we explore different ratios of subsampling for half of the classes. The subsampling rates are 80% in the moderate and 60% in the severe case. In Table 2 we evaluate the methods with respect to their clustering accuracy (Acc), normalized mutual information (NMI) and adjusted rand index (ARI).

The hypergeometric prior distribution shows fairly good clustering performance in all datasets. Although the uniform distribution performs reasonably well, it assumes the clusters have equal importance. Hence it might fail in more complex scenarios. On the other hand, the categorical distribution has subpar performance compared to the uniform distribution, even in the moderate subsampling setting. This might be due to the additional complexity given by the learnable cluster weights, which results in unstable results. On the contrary, the additional complexity does not seem to affect the performance of the proposed hypergeometric prior but instead boosts its clustering performance, especially in the imbalanced dataset. In Figure 3, we show that the model is able to learn the weights, $\boldsymbol{\pi}$, which reflect the subsampling rates of each cluster, which is not directly possible using the uniform prior model.

## 6 CONCLUSION

We propose the differentiable hypergeometric distribution in this work. In combination with the Gumbel-Softmax trick, this new formulation enables reparametrized gradients with respect to the class weights $\boldsymbol{\omega}$ of the hypergeometric distribution. We show the various possibilities of the hypergeometric distribution in machine learning by applying it to two common areas, clustering, and weakly-supervised learning. In both applications, methods using the hypergeometric distribution reach at least state-of-the-art performance. We believe this work is an essential step toward integrating the hypergeometric distribution into more machine learning algorithms. Applications in biology and social sciences represent potential directions for future work.

ACKNOWLEDGMENTS

We would like to thank Ričards Marcinkevičs for good discussions and help with the Kolmogorov-Smirnov-Test. TS is supported by the grant #2021-911 of the Strategic Focal Area "Personalized

Health and Related Technologies (PHRT)" of the ETH Domain (Swiss Federal Institutes of Technology). LM is supported by the SDSC PhD Fellowship #1-001568-037. AR is supported by the StimuLoop grant #1-007811-002 and the Vontobel Foundation.

## 7 Ethics Statement

In this work, we propose a general approach to learning the importance of subgroups. In that regard, potential ethical concerns arise with the different applications our method could be applied to. We intend to apply our model in the medical domain in future work. Being able to correctly model the dependencies of groups of patients is important and offers the potential of correctly identifying underlying causes of diseases on a group level. On the other hand, grouping patients needs to be handled carefully and further research is needed to ensure fairness and reliability with respect to underlying and hidden attributes of different groups.

## 8 Reproducibility Statement

For all theoretical statements, we provide detailed derivations and state the necessary assumptions. We present empirical support on both synthetic and real data to back our idea of introducing the differentiable hypergeometric distribution. To ensure empirical reproducibility, the results of each experiment and every ablation were averaged over multiple seeds and are reported with standard deviations. All of the used datasets are public or can be generated from publicly available resources using the code that we provide in the supplementary material. Information about implementation details, hyperparameter settings, and evaluation metrics are provided in the main text or the appendix.

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

# A  PRELIMINARIES

## A.1  HYPERGEOMETRIC DISTRIBUTION

The hypergeometric distribution is a discrete probability distribution that describes the probability of $x$ successes in $n$ draws without replacement from a finite population of size $N$ with $m$ elements that are part of the success class Unlike the binomial distribution, which describes the probability distribution of $x$ successes in $n$ draws with replacement.

**Definition A.1** (Hypergeometric Distribution (Gonin, 1936)[4])**.** A random variable $X$ follows the hypergeometric distribution, if its probability mass function (PMF) is given by

$$P(X = x) = p_X(x) = \frac{\binom{m}{x}\binom{N-m}{n-x}}{\binom{N}{n}} \tag{9}$$

---

[4]Although the distribution itself is older, Gonin (1936) were the first to name it hypergeometric distribution

Urn models are typical examples of the hypergeometric probability distribution. Suppose we think of an urn with marbles in two different colors, e. g. green and purple, we can label as success the drawing of a green marble. Then $N$ defines the total number of marbles and $m$ the number of green marbles in the urn. $x$ is the number of green marbles, and $n - x$ is the number of drawn purple marbles.

The multivariate hypergeometric distribution describes an urn with more than two colors, e.g. green, purple and yellow in the simplest case with three colors. As described in Johnson (1987), the definition is given by:

**Definition A.2** (Multivariate Hypergeometric Distribution)**.** A random vector $\boldsymbol{X}$ follows the multivariate hypergeometric distribution, if its joint probability mass function is given by

$$P(\boldsymbol{X} = \boldsymbol{x}) = p_{\boldsymbol{X}}(\boldsymbol{x}) = \frac{\prod_{i=1}^{c} \binom{m_i}{x_i}}{\binom{N}{n}} \tag{10}$$

where $c \in \mathbb{N}_+$ is the number of different classes (e.g. marble colors in the urn), $\boldsymbol{m} = [m_1, \ldots, m_c] \in \mathbb{N}^c$ describes the number of elements per class (e.g. marbles per color), $N = \sum_{i=1}^{c} m_i$ is the total number of elements (e.g. all marbles in the urn) and $n \in \{0, \ldots, N\}$ is the number of elements (e.g. marbles) to draw. The support $\mathcal{S}$ of the PMF is given by

$$\mathcal{S} = \left\{ \boldsymbol{x} \in \mathbb{N}_0^c : \forall i \quad x_i \leq m_i, \sum_{i=1}^{c} x_i = n \right\} \tag{11}$$

### A.2 GUMBEL-SOFTMAX-TRICK

Most of the information in this section is from (Jang et al., 2016; Maddison et al., 2017), which concurrently introduced the Gumbel-Softmax trick. Gumbel-Softmax is a continuous distribution that can approximate the categorical distribution, and whose parameter gradients can be easily computed using the reparameterization trick.

Let $\boldsymbol{z}$ be a categorical variable with categorical weights $\boldsymbol{\pi} = [\pi_1, \ldots, \pi_C]$ such that $\sum_{k=1}^{C} \pi_k = 1$. Following (Jang et al., 2016), we assume that categorical samples are encoded as one-hot vectors. The Gumbel-Max trick (Gumbel, 1954; Maddison et al., 2014) defines an efficient way to draw samples $\boldsymbol{z}$ from a categorical distribution with weights $\boldsymbol{\pi}$:

$$\boldsymbol{z} = \text{one\_hot}(\arg\max_k \log(\pi_k) + g_k) \tag{12}$$

where $\boldsymbol{g} = [g_1, \ldots, g_C]$ are i. i. d. samples drawn from Gumbel(0,1). We can efficiently sample $g$ from Gumbel(0,1) by drawing a sample $u$ from a uniform distribution $U(0,1)$ and applying the transform $g = -\log(-\log u)$. For more details, we refer to (Gumbel, 1954; Maddison et al., 2014).

(Jang et al., 2016; Maddison et al., 2017) both use the softmax function as a continuous and differentiable approximation to $\arg\max$. The softmax function is defined as

$$p_k = \frac{\exp((\log \pi_k + g_k)/\tau)}{\sum_{j=1}^{C} \exp((\log \pi_j + g_j)/\tau)} \quad \text{for} \quad k = 1, \ldots, C \tag{13}$$

where $\tau$ is a temperature parameter. As $\tau$ goes to zero, the softmax function approximates the argmax function. Hence, the Gumbel-Softmax distribution approximates the categorical distribution.

## B METHODS

### B.1 PMF FOR THE MULTIVARIATE FISHER'S NONCENTRAL DISTRIBUTION

In this section, we give a detailed derivation for the calculation of the log-probabilities of the multivariate Fisher's noncentral hypergeometric distribution. We end up with a formulation that is proportional to the actual log-probabilities. Because the ordering of categories is not influenced by scaling with a constant factor (addition/subtraction in log domain), these are unnormalized

log-probabilities of the multivariate Fisher's noncentral hypergeometric distribution.

$$p_{\boldsymbol{X}}(\boldsymbol{x}; \boldsymbol{\omega}) = \frac{1}{P_0} \prod_{i=1}^{c} \binom{m_i}{x_i} \omega_i^{x_i} \tag{14}$$

where $P_0$ is defined as in Equation (1). From there it follows

$$\log p_{\boldsymbol{X}}(\boldsymbol{x}; \boldsymbol{\omega}) = \log \left( \frac{1}{P_0} \prod_{i=1}^{c} \binom{m_i}{x_i} \omega_i^{x_i} \right) \tag{15}$$

$$= \log \left( \frac{1}{P_0} \right) + \log \left( \prod_{i=1}^{c} \binom{m_i}{x_i} \omega_i^{x_i} \right) \tag{16}$$

$$= \log \left( \frac{1}{P_0} \right) + \sum_{i=1}^{c} \log \left( \binom{m_i}{x_i} \omega_i^{x_i} \right) \tag{17}$$

$$= \log \left( \frac{1}{P_0} \right) + \sum_{i=1}^{c} \left( \log \binom{m_i}{x_i} + \log \left( \omega_i^{x_i} \right) \right) \tag{18}$$

$$= \log \left( \frac{1}{P_0} \right) + \sum_{i=1}^{c} \left( \log \binom{m_i}{x_i} + x_i \log \left( \omega_i \right) \right) \tag{19}$$

Constant factor can be removed as the argmax is invariant to scaling with a constant factor which equals addition or subtraction in $\log$-space. It follows

$$\log p_{\boldsymbol{X}}(\boldsymbol{x}; \boldsymbol{\omega}) = \sum_{i=1}^{c} \left( \log \binom{m_i}{x_i} + x_i \log \left( \omega_i \right) \right) + C \tag{20}$$

$$= \sum_{i=1}^{c} \left( \log \frac{1}{x_i!(m_i - x_i)!} + x_i \log \left( \omega_i \right) \right) + \tilde{C} \tag{21}$$

$$= \sum_{i=1}^{c} \left( - \log \left( \Gamma(x_i + 1)\Gamma(m_i - x_i + 1) \right) + x_i \log \left( \omega_i \right) \right) + \tilde{C} \tag{22}$$

$$\tag{23}$$

In the last line we used the relation $\Gamma(k + 1) = k!$. Setting $C = \tilde{C}$, it directly follows

$$\log p_{\boldsymbol{X}}(\boldsymbol{x}; \boldsymbol{\omega}) = \sum_{i=1}^{c} x_i \log \omega_i + \psi_F(\boldsymbol{x}) + C \tag{24}$$

where $\psi_F(\boldsymbol{x}) = - \sum_{i=1}^{c} \log \left( \Gamma(x_i + 1)\Gamma(m_i - x_i + 1) \right)$.

The Gamma function is defined in Whittaker & Watson (1996) as

$$\Gamma(z) = \int_0^{\infty} x^{z-1} e^{-x} dx \tag{25}$$

### B.2 Proof for Lemma 4.1

*Proof.* Factors that are constant for all $x$ do not change the relative ordering between different values of $x$. Hence, removing them preserves the ordering of values $x$ (Barrett, 2017).

$$\log p_{X_L}(x_L; \boldsymbol{\omega}) = \log \left( \frac{1}{P_0} \binom{m_L}{x} \omega_L^{x_L} \binom{m_R}{n - x_L} \omega_R^{n-x_L} \right) \tag{26}$$

$$= \log \binom{m_L}{x_L} + \log \binom{m_R}{n - x_L} + \log \left( \omega_L^{x_L} \right) + \log \left( \omega_R^{n-x_L} \right) + C \tag{27}$$

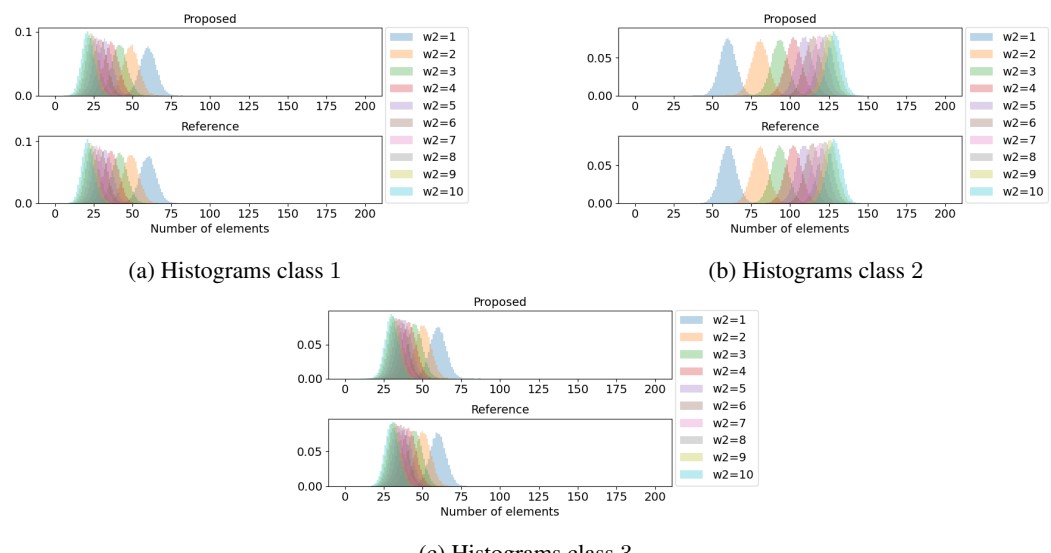

(a) Histograms class 1               (b) Histograms class 2

(c) Histograms class 3

Figure 4: Comparing random variables drawn from the proposed distribution to a reference distribution.

Using the definition of the binomial coefficient (see Section 3) and its relation to the Gamma function[5] $\Gamma(k+1) = k!$, it follows

$$
\begin{aligned}
\log p_{X_L}(x_L; \boldsymbol{\omega}) =\; & x \cdot \log \omega_L + (n - x_L) \cdot \log \omega_R \\
& - \log\left(\Gamma(x_L + 1)\Gamma(n - x_L + 1)\right) \\
& - \log\left(\Gamma(m_L - x_L + 1)\Gamma(m_R - n + x_L + 1)\right) + C
\end{aligned}
\tag{28}
$$

With $\psi_F(x)$ as defined in Equation (6), Equation (5) follows directly. $\qquad\square$

### B.3    PROOF FOR LEMMA 4.2

*Proof.* When sampling class $i$, we draw $x_i$ samples from class $i$ where $x_i \leq m_i$. The conditional distribution $p_{X_i}(x_i | \{x_k\}_{k=1}^{i-1}; \boldsymbol{\omega})$ for class $i$ given the already sampled classes $k < i$ simultaneously defines the weights of a categorical distribution. Sampling $x_i$ elements from class $i$ can be seen as selecting the $x_i$th category from the distribution defined by the weights $p_{X_i}(x_i | \{x_k\}_{k=1}^{i-1}; \boldsymbol{\omega})$. Therefore,

$$
\sum_{0 \leq x_i \leq m_i} p_{X_i}\left(x_i | \{x_k\}_{k=1}^{i-1}; \boldsymbol{\omega}\right) = 1,
\tag{29}
$$

which allows us to apply the Gumbel-Max trick and, respectively, the Gumbel-Softmax trick. $\qquad\square$

### B.4    THOUGHT EXPERIMENT FOR MODELING THE HYPERGEOMETRIC DISTRIBUTION WITH A SEQUENCE OF UNCONSTRAINED CATEGORICAL DISTRIBUTIONS

We quickly touched on the topic of using a sequence of categorical distributions in the main text (Section 4.3). To further describe and discuss the problem of using a sequence of categorical distributions, we provide a more detailed explanation and example here. Of course, the same constraints as the ones described in the main paper apply, e. g. we want our method to be differentiable, scalable in the number of random states and computationally efficient. Hence, we are interested in methods that infer at least all states of a single class in parallel, and not in methods that sequentially iterate over all possible random states for every re-sampling operation.

---

[5]see Appendix B.1 for the definition on the Gamma function

In our example, we want to model a hypergeometric distribution with 3 classes and the following specifications

$$m_1 = 10, \quad m_2 = 7, \quad m_3 = 8, \quad n = 9 \tag{30}$$

We use one categorical distribution for every class. The categories of the categorical distributions describe the number of elements to sample from this class. Based on the three categorical distributions, we would like to sample $x_j \in \mathbb{N}_0$ such that $\sum_j x_j = n$. The sequence of distributions ideally should describe the class conditional distributions of the hypergeometric distribution described in Section 4.1. Or at least model sampling without replacement, i. e. fulfill the necessary constraint.

We then have three vectors $\boldsymbol{\pi}_1$, $\boldsymbol{\pi}_2$ and $\boldsymbol{\pi}_3$ which define the categorical weights of the three categorical distributions. As such $\sum_k \pi_{j,k} = 1 \quad \forall \quad j$. Using three categorical distributions, we are not able to explicitly model $\boldsymbol{\omega}$, but if we are able to fulfill the constraints, there would be some matching $\boldsymbol{\omega}$. To make it differentiable, we approximate the categorical distributions using the Gumbel-Softmax (GS) trick, such that we can use everything in a differentiable pipeline. In most differentiable settings, the categorical weights are inferred using some neural network, e. g.

$$\boldsymbol{\pi}_i = f_{\xi,i}(\cdot) \tag{31}$$

Knowing that our categories correspond to integer values, we can use the straight-through estimator (Bengio et al., 2013) together with the Gumbel-Softmax trick and a bit of matrix multiplication to convert an one-hot vector to an integer value:

$$\boldsymbol{y}_j = \text{straight\_through}(GS(\boldsymbol{\pi}_j)) \tag{32}$$
$$= \arg\max(GS(\boldsymbol{\pi}_j)) \tag{33}$$

Please check the provided code for more details on the details of the matrix multiplications.

We can either infer the categorical weights of and sample from the three distributions in parallel or sequentially. We first try to infer the weights of all distributions in parallel. With the constraint $\sum_j x_j = n$, not all combinations of $\boldsymbol{\pi}_j, \forall j$ are valid anymore. We might be able to learn the constraints, such that the number of samples fulfilling them is maximized. But it is not possible to have the constraints respected due to the formulation of this method. Hence, it is not guaranteed that we sample $x_j, \forall j$ such that $\sum_j x_j = n$, which then results in non-valid samples.

Therefore, we try to infer the $\boldsymbol{\pi}_j$ sequentially. The sequential procedure models the following behaviour (similar to our proposed class conditional sampling)

$$p(x_1, x_2, x_3) = p(x_1)p(x_2 \mid x_1)p(x_3 \mid x_1, x_2) \tag{34}$$

Without loss of generality, we assume that we first infer $\boldsymbol{\pi}_1$ and sample $x_1 \sim Cat(\boldsymbol{\pi}_1)$ such that $x_1 = 7$. It follows that not all combination of weights $\boldsymbol{\pi}_2$ and $\boldsymbol{\pi}_3$ are valid anymore. When inferring $\boldsymbol{\pi}_2$ some weights have to be zero to guarantee that $\sum_j x_j = n$ and sample a valid random vector. Hence,

$$\pi_{2,k} = 0 \quad \forall \quad k > 2 \tag{35}$$

If we assign any nonzero probability to $\pi_{2,k}, k > 2$, we are not able to fulfill the constraint $\sum_j x_j = n$ anymore for every generated sample. Additionally, from Equation (12) it follows that we would need to constrain the gumbel noise as well. We are unaware of previous work that proposed a constrained Gumbel-Softmax trick which would model this behaviour. Also, there is no guarantee that $\pi_{2,k} > 0, k \leq 2$, leading to additional heuristics-based solution that we would need to implement. There is an unclear effect on the calculation of the gradients, which would need to be investigated. Also, for sampling and inferring of weights $\boldsymbol{\pi}_j$, there arise questions of ordering between classes and how valid random samples can be guaranteed. We summarize that the implementation and modeling of a hypergeometric distribution using a sequence of unconstrained categorical distributions is far from being trivial, because of the open question of how implement constraints in a general and dynamic way when using the Gumbel-Softmax trick.

Note the important difference to our hypergeometric distribution. Although we also use the Gumbel-Softmax trick to generate random samples, we are able to infer $\boldsymbol{\omega}$ in parallel, which does not introduce an implicit ordering between classes and the constraint $\sum_j x_j = n$ is guaranteed. The only constraint with respect to $\boldsymbol{\omega}$ we have to satisfy, is $\omega_j \geq 0, \forall j$. $\omega_j \geq 0$ can easily be satisfied using a ReLU

---

**Algorithm 2** Subroutines for sampling From Multivariate Noncentral Hypergeometric Distribution.

**function** SAMPLEUNCHG($m_i, m_j, \omega_i, \omega_j, n, \tau$)
    $\boldsymbol{\alpha}_i \leftarrow \text{calcLogPMF}(m_i, m_j, \omega_i, \omega_j, n)$            # Section 4.2
    $x_i, \hat{\boldsymbol{r}}_i \leftarrow \text{contRelaxSample}(\boldsymbol{\alpha}_i, \tau))$           # Section 4.3
    **return** $x_i, \boldsymbol{\alpha}_i, \hat{\boldsymbol{r}}_i$
**end function**

**function** CALCLOGPMF($m_l, m_r, \omega_l, \omega_r, n$)
    **for** $k \in \{0, \ldots, m_l\}$ **do**
        $\boldsymbol{x}_{l,k} \leftarrow (k + 1)$
        $\boldsymbol{x}_{r,k} \leftarrow (\text{ReLU}(n - k) + 1)$
    **end for**
    $\boldsymbol{l} \leftarrow \log \Gamma(\boldsymbol{x}_l + 1) + \log \Gamma(m_l - \boldsymbol{x_l} + 1)$       # see Appendix B.5
    $\boldsymbol{r} \leftarrow \log \Gamma(\boldsymbol{x}_r + 1) + \log \Gamma(m_r - \boldsymbol{x_r} + 1)$
    $\boldsymbol{\alpha}_l \leftarrow \boldsymbol{x}_l \log \omega_l + \boldsymbol{x}_r \log \omega_r - (\boldsymbol{l} + \boldsymbol{r})$
    **return** $\boldsymbol{\alpha}_l$
**end function**

**function** CONTRELAXSAMPLE($\boldsymbol{\alpha}_l, \tau$)
    $\boldsymbol{u} \leftarrow \boldsymbol{U}(\boldsymbol{0}, \boldsymbol{1})$
    $\boldsymbol{g} \leftarrow -\log(-\log \boldsymbol{u})$
    $\hat{\boldsymbol{r}}_l \leftarrow \boldsymbol{\alpha}_l + \boldsymbol{g}$
    $\boldsymbol{p}_l \leftarrow \text{Softmax}(\hat{\boldsymbol{r}}_l / \tau)$
    $x_l \leftarrow \text{Count-Index}(\text{Straight-Through}(\boldsymbol{p}_l))$     # Count-Index and Straight-Through:
                                                           # see Appendix B.5
    **return** $x_l, \hat{\boldsymbol{r}}_l$
**end function**

---

activation function. We are then able to calculate the class conditional distributions as a function of $\boldsymbol{\omega}$ such that all constraints are satisfied. We do not need to dynamically change categorical weights, i. e. model parameters, during the sampling procedure, which is the case for the sequence of constrained categorical distributions.

## B.5 ALGORITHM AND MINIMAL EXAMPLE

In the main text, we drafted the pseudocode for our proposed algorithm (see Algorithm 1). We only did it for the main functionality, but not the subroutines described in Sections 4.2 and 4.3). Algorithm 2 describes the subroutines used in Algorithm 1 and explained in Section 4.

The straight-through operator uses a hard one-hot embedding in the forward path, but the relaxed vector for the backward path and the calculation of the derivative (Bengio et al., 2013). The Count-Index maps the one-hot vector to an index, which is equal to the number of selected class elements in our case.

### B.5.1 MINIMAL EXAMPLE

We describe a minimal example application using a step-by-step procedure to provide intuition and further illustrate the proposed method. Here, we learn the generative model of an urn model using stochastic gradient descent when given samples from an urn model with a priori unknown weights $\boldsymbol{\omega}$. Note that given a dataset, we could also estimate the weights $\boldsymbol{\omega}$ of the hypergeometric distribution by minimizing the negative log probability of the data given the parameters $\log p_X(\boldsymbol{x}; \boldsymbol{\omega})$. In contrast, we use a generative approach to demonstrate how our method allows backpropagation when modeling the generative process of the samples by reparameterizing the hypergeometric distribution. Additionally, we illustrate the minimal example with two figures (Figures 5 and 6), which explain the sampling procedure visually.

Let us start with our minimal example. We are given a dataset of *i.i.d* samples $\boldsymbol{X}_D \in \mathbb{N}_+^{K \times n_c}$ from a multivariate noncentral distribution with unknown $\boldsymbol{\omega}_{gt} \in \mathbb{R}_+^c$. $K$ is the number of samples in the

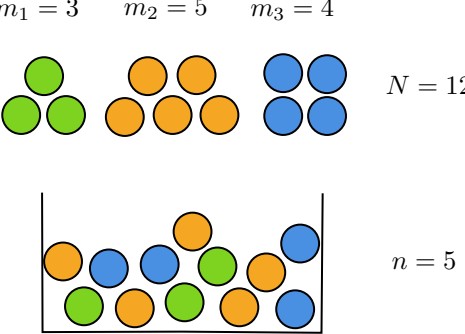

Figure 5: Illustration of the basic setting of the multivariate hypergeometric distribution. We have 3 classes of elements (green, orange, and blue) with different and unknown importance $\omega_c$ for every class $c$. In our urn, the total number of elements $N$ is given by the sum of elements of all classes $m_c$. From this urn, we draw a group of $n$ samples. In this example, $n = 5$. The group importance $\omega_c$ is often unknown, and difficult to estimate. Our formulation helps to learn $\omega_c$ using gradient-based optimization when simulating how given samples are drawn from the urn.

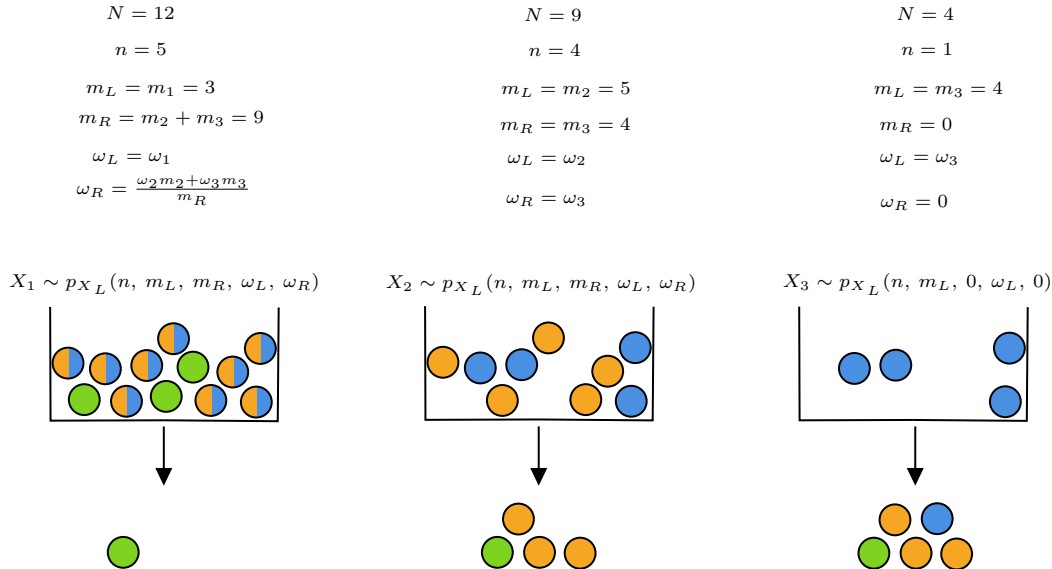

Figure 6: Illustration of the proposed conditional sampling from the multivariate noncentral hypergeometric distribution. We use the same urn as in Figure 5 with $\boldsymbol{m} = [3, 5, 4]$ and $n = 5$. As described in Section 4, we sequentially sample random variates of the individual classes. Hence, we start by sampling class 1. For that, we merge classes 2 and 3 (illustrated by the half blue and half orange balls) creating the necessary parameters $m_L, m_R, \omega_L, \omega_R$ for $p_{X_L}(\cdot)$ (described in the left column). This is also described in Algorithms 1 and 2. Using the univariate distribution $p_{X_L}(\cdot)$ we sample the random variable $X_1$, which is equal to 1 in our example (symbolized by the single green ball). We continue with sampling the class 2, which is described in the middle column. The merge operation simplifies to assigning $m_L = m_2$ and $m_R = m_3$, and $n$ is the original $n$ minus $X_1$. We draw $X_2 = 3$ in our example (again illustrated by the 3 orange balls below the urn). Because the number of drawn balls must sum to $n$, the last class $X_3$ is fully determined already.

dataset and $n_c$ defines the number of classes. A sample from the dataset $\boldsymbol{X}_D$ is denoted as $X_D$. For every $X_D \in \boldsymbol{X}_D$, it holds that $\sum_c X_{D,c} = n$. We assume that we know the total number of elements in the urn, e.g. $\boldsymbol{m} = [m_1, m_2, ..., m_c]$.

In our minimal example, we are interested in learning the unknown importance weights $\boldsymbol{\omega}$ with a generative model using stochastic gradient descent (SGD). Hence, we assume a data generating distribution $p_X(\boldsymbol{x}; \boldsymbol{\omega})$ such that $X \sim p_X(\boldsymbol{x}; \boldsymbol{\omega})$. The loss function $\mathbb{L}$ is given as

$$\mathbb{L} = \sum_{X_D \in \boldsymbol{X}_D} \mathbb{E}_{X \sim p_X(\boldsymbol{x}; \boldsymbol{\omega})} \left[ (X_D - X)^2 \right] \tag{36}$$

$$= \sum_{X_D \in \boldsymbol{X}_D} \mathbb{E}_{X \sim p_X(\boldsymbol{x}; \boldsymbol{\omega})} \left[ \mathbb{L}(X_D, X) \right] \tag{37}$$

where $\mathbb{L}$ is the loss per sample. $p_X(\boldsymbol{x}; \boldsymbol{\omega})$ is a noncentral multivariate hypergeometric distribution as defined in Definition 3.1 where the class importance $\boldsymbol{\omega}$ is unknown.

To minimize $\mathbb{E}[\mathbb{L}(X)]$, we want to optimize $\boldsymbol{\omega}$. Using SGD, we optimize the parameters $\boldsymbol{\omega}$ in an iterative manner:

$$\boldsymbol{\omega}_{t+1} := \boldsymbol{\omega}_t - \eta \frac{d}{d\boldsymbol{\omega}} \mathbb{E}_{X \sim p_X(\boldsymbol{x}; \boldsymbol{\omega})} \left[ \mathbb{L}(X_D, X) \right] \tag{38}$$

where $\eta$ is the learning rate, and $t$ is the step in the optimization process. Unfortunately, we do not have a reparameterization estimator $\frac{d}{d\boldsymbol{\omega}} \mathbb{E}[\mathbb{L}(X_D, X)]$ because of the jump discontinuities of the $\arg\max$ function in the categorical distributions.

As described in Sections 4.1 and 4.3, we can rewrite $p_X(\boldsymbol{x}; \boldsymbol{\omega})$ as a sequence of conditional distributions. Every conditional distribution is itself a categorical distribution, which prohibits the calculation of $\frac{d}{d\boldsymbol{\omega}} \mathbb{E}[\mathbb{L}(X_D, X)]$.

In more detail, we rewrite the joint probability distribution $p_X(\boldsymbol{x}; \boldsymbol{\omega})$ as

$$p_X(\boldsymbol{x}; \boldsymbol{\omega}) = p_{X_1}(x_1; \boldsymbol{\omega}) \prod_{c=2}^{n_c} p_{X_c}(x_c \mid x_1, ..., x_{c-1}; \boldsymbol{\omega}) \tag{39}$$

where every distribution $p_{X_c}(\cdot; \boldsymbol{\omega})$ is a categorical distribution. We sample every $X_c$ using Equation (5), i.e.

$$p_{X_c}(x_{L_c}; \boldsymbol{\omega}) = x_{L_c} \log \omega_{L_c} + (n_c - x_L) \log \omega_{R_c} + \psi_F(x_{L_c}) + C \tag{40}$$

$\omega_{L_c}, \omega_{R_c}, m_{L_c}, m_{R_c}$, and $n_c = \sum_{j<c} X_j$ are calculated according to Equation (3) and sequentially for every class.

The expected element-wise loss $\mathbb{E}_{X \sim p_X(\boldsymbol{x}; \boldsymbol{\omega})} \left[ \mathbb{L}(X_D, X) \right]$ changes to

$$\mathbb{E}_{X \sim p_X(\boldsymbol{x}; \boldsymbol{\omega})} \left[ \mathbb{L}(X_D, X) \right] = \mathbb{E}_{X \sim p_X(\boldsymbol{x}; \boldsymbol{\omega})} \left[ \sum_{c=1}^{n_c} (X_{D,c} - X_c)^2 \right] \tag{41}$$

$$= \mathbb{E}_{X \sim p_X(\boldsymbol{x}; \boldsymbol{\omega})} \left[ \sum_{c=1}^{n_c} \mathbb{L}(X_{D,c}, X_c) \right] \tag{42}$$

$$= \sum_{c=1}^{n_c} \mathbb{E}_{X \sim p_X(\boldsymbol{x}; \boldsymbol{\omega})} \left[ \mathbb{L}(X_{D,c}, X_c) \right] \tag{43}$$

Hence,

$$\frac{d}{d\boldsymbol{\omega}} \mathbb{E}[\mathbb{L}(X_D, X)] = \sum_{c=1}^{n_c} \frac{d}{d\boldsymbol{\omega}} \mathbb{E}_{X \sim p_X(\boldsymbol{x}; \boldsymbol{\omega})} \left[ \mathbb{L}(X_{D,c}, X_c) \right] \tag{44}$$

Unfortunately, for every $\frac{d}{d\boldsymbol{\omega}} \mathbb{E}\left[ \mathbb{L}(X_{D,c}, X_c) \right]$, we face the problem of not having a reparameterizable gradient estimator. We cannot calculate the gradients of the loss directly, but $p_{X_c}(\cdot)$ being categorical distributions allows us to use the Gumbel-Softmax gradient estimator (Jang et al., 2016; Maddison et al., 2014; Paulus et al., 2020).

The Gumbel-Softmax trick is a differentiable approximation to the Gumbel-Max trick (Maddison et al., 2014), which provides a simple and efficient way to draw samples from a categorical distribution with weights $\boldsymbol{\pi}$. The Gumbel-Softmax trick uses the softmax function as a differentiable approximation to the `argmax` function used in the Gumbel-Max trick. It follows (Jang et al., 2016)

$$\boldsymbol{y} = \text{softmax}((\log \boldsymbol{\pi} + \boldsymbol{g})/\tau) \tag{45}$$
$$= \text{softmax}_\tau(\log \boldsymbol{\pi} + \boldsymbol{g}) \tag{46}$$

where $g_1, \ldots, g_k$ are *i.i.d.* samples drawn from Gumbel$(0, 1)$, and $\tau$ is a temperature parameter. $\boldsymbol{y}$ is a continuous approximation to a one-hot vector, i.e. $0 \le y_i \le 1$ such that $\sum_i y_i = 1$.

Different to the standard Gumbel-Softmax trick, we infer the weights $\boldsymbol{\pi}$ from the probability density function $\log p_X(\cdot)$ (see Equations (7) and (8)). We write for a single conditional class $x_{L_c}$ as the procedure is the same for all classes. It follows

$$X_{\tau,c}(\boldsymbol{\omega}, \boldsymbol{g}) = \text{softmax}_\tau(\log p_{X_c}(\boldsymbol{x}_{L_c}; \boldsymbol{\omega}) + \boldsymbol{g}) \tag{47}$$

where $\boldsymbol{x}_{L_c} = [0, m_{L_c}]$. Due to the translation invariance of the softmax function, we do not need to calculate the constant $C$ in $\log p_{X_c}(\boldsymbol{x}_{L_c}; \boldsymbol{\omega})$.

The Gumbel-Softmax approximation is smooth for $\tau > 0$, and therefore $\mathbb{E}[\mathbb{L}(X_D, X_\tau)]$ has well-defined gradients $\frac{d}{d\boldsymbol{\omega}}$.

We write the loss function to optimize and its gradients as

$$\mathbb{E}_{\boldsymbol{g}}\left[\mathbb{L}(X_D, X_\tau(\boldsymbol{\omega}, \boldsymbol{g}))\right] = \sum_{c=1}^{n_c} \mathbb{E}_{\boldsymbol{g}}\left[\mathbb{L}(X_{D,c}, X_{\tau,c}(\boldsymbol{\omega}, \boldsymbol{g}))\right] \tag{48}$$

$$\frac{d}{d\boldsymbol{\omega}}\mathbb{E}_{\boldsymbol{g}}\left[\mathbb{L}(X_D, X_\tau(\boldsymbol{\omega}, \boldsymbol{g}))\right] = \sum_{c=1}^{n_c} \mathbb{E}_{\boldsymbol{g}}\left[\frac{d}{d\boldsymbol{\omega}}\mathbb{L}(X_{D,c}, X_{\tau,c}(\boldsymbol{\omega}, \boldsymbol{g}))\right] \tag{49}$$

By replacing the categorical distribution in Equation (5) with the Gumbel-Softmax distribution (see Lemma 4.2), we can thus use backpropagation and automatic differentiation frameworks to compute gradients and optimize the parameters $\boldsymbol{\omega}$ (Jang et al., 2016).

We implemented our minimal example for $n_c = 3$ classes. We set $\boldsymbol{m} = [m_1, m_2, m_3] = [200, 200, 200]$ and $n = 180$. We create 10 datasets $\boldsymbol{X}_D \in \mathbb{N}^{1000 \times 3}$ generated from different $\boldsymbol{\omega}_{gt}$ and show the performance of the proposed method. From these 1000 samples, we use 800 for training and 200 for validation. Similar to the setting we use for the KS-test (see Section 5.1), we choose 10 values for $\omega_{gt,2}$, i.e. $\omega_{gt,2} = [1.0, 2.0, \ldots, 10.0]$. The values for $\omega_{gt,1}$ and $\omega_{gt,3}$ are set to 1.0 for all datasets versions.

As described above the model does not have access to the data generating true $\boldsymbol{\omega}_{gt}$. So, for every dataset $\boldsymbol{X}_D$, we optimize the unknown $\boldsymbol{\omega}$ based on the loss $\mathbb{L}$ defined in Equation (43).

Figure 7 shows the training and validation losses over the training steps. We train the model for 10 epochs, but we see that the model converges earlier. The losses only differ at the beginning of the training procedure, which is probably an initialization effect, but quickly converge to similar values independent of the $\omega_2$ value that generated the dataset. Figure 8 shows the estimation of $\log \boldsymbol{\omega}$. The $x$-axis shows the training step, the $y$-axis shows the estimated value. Figures 8a to 8b demonstrate that the hypergeometric distribution is invariant to the scale of $\boldsymbol{\omega}$. With increasing value of $\omega_{gt,2}$, the values of $\omega_1$ and $\omega_3$ decrease, although their ground truth values $\omega_{gt,1}$ and $\omega_{gt,3}$ do not change. The training and validation loss do not increase though (Figures 7a and 7b), which demonstrates the scale-invariance.

## C EXPERIMENTS

All our experiments were performed on our internal compute cluster, equipped with NVIDIA RTX 2080 and NVIDIA RTX 1080. Every training and test run used only a single NVIDIA RTX 2080 or NVIDIA RTX 1080. The weakly-supervised runs took approximately 3-4 hours each, where the clustering runs only took about 3 hours each. We report detailed runtimes for the weakly-supervised experiments in Table 3 to highlight the efficiency of the proposed method.

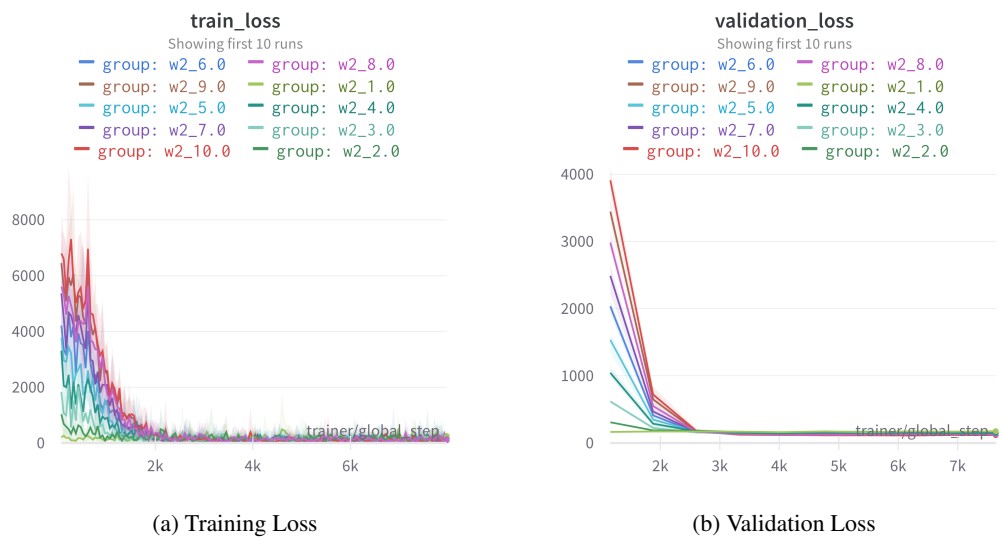

(a) Training Loss

(b) Validation Loss

Figure 7: Training and validation losses for different values of $\boldsymbol{\omega}_{gt}$ of our minimal example described in Appendix B.5.1.

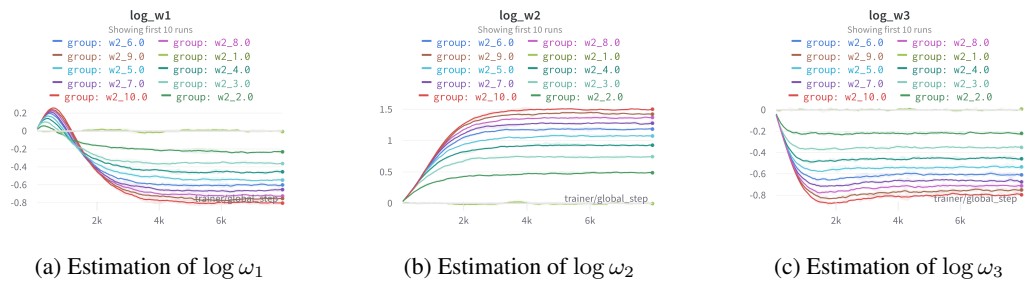

(a) Estimation of $\log \omega_1$

(b) Estimation of $\log \omega_2$

(c) Estimation of $\log \omega_3$

Figure 8: The estimated $\log \boldsymbol{\omega}$ values over training procedure for different ground truth $\boldsymbol{\omega}_{gt}$ values of our minimal example (see Appendix B.5.1). We see nicely that the hypergeometric distribution is invariant to the scale of $\boldsymbol{\omega}$. With increasing value of $\omega_2$, the estimated values for $\omega_1$ and $\omega_3$ change as well, but the training and validation loss remain low (see Figure 7).

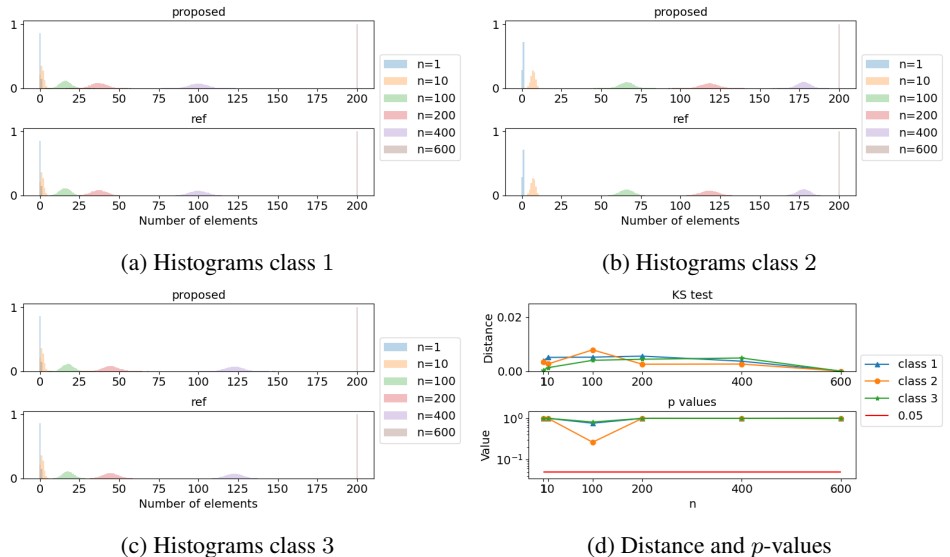

Figure 9: Sensitivity analysis for a varying number of samples to draw $n$ measured with the Kolmogorov-Smirnov test. In this experiment, we have the following specifications: $\boldsymbol{m} = \{200, 200, 200\}$, $\boldsymbol{\omega} = \{1.0, 5.0, 1.0\}$. The values of $n$ are given in the plot.

### C.1 KOLMOGOROV-SMIRNOV TEST

As described in Section 5.1 in the main text, we report the histograms of all classes next to each other here in Figure 4. We can see that class 1 and class 3 have (at least visually) the same distribution over histograms for different values of $\omega_2$. And they also match their respective reference histograms.

#### C.1.1 ABLATION STUDIES FOR KOLMOGOROV-SMIRNOV TEST

To highlight our accurate approximation, we provide more results from KS-tests here in the appendix. We perform additional tests with varying $n$ and $m_2$. See Figures 9 and 10 for the detailed results. We see that over all combinations, our approximation of the hypergeometric distribution performs well and produces samples of approximately the same quality as the reference distribution.

### C.2 WEAKLY-SUPERVISED LEARNING

#### C.2.1 METHOD, IMPLEMENTATION AND HYPERPARAMETERS

In this section we give more details on the used methods. We make use of the `disentanglement_lib` (Locatello et al., 2019) which is also used in the original paper we compare to (Locatello et al., 2020). The baseline algorithms (Bouchacourt et al., 2018; Hosoya, 2018) are already implemented in `disentanglement_lib`. For details on the implementation of models, we refer to the original paper. We did not change any hyperparameters or network settings. All experiments were performed using $\beta = 1.0$ as this is the best performing $\beta$ according to Locatello et al. (2020). For all experiments we train three models with different random seeds.

All experiments are performed using GroupVAE (Hosoya, 2018). Using GroupVAE, shared latent factors are aggregated using an arithmetic mean. Bouchacourt et al. (2018) assume also knowledge about shared and independent latent factors. In contrast to GroupVAE, their ML-VAE aggregates shared latent factors by using the Product of Experts (i.e. geometric mean).

Figure 12 shows the basic architecture. The different architectures only differ in the `View Aggregation` module. In this module, every method selects the latent factors $z_i \in S$, which should be aggregated over different views $\boldsymbol{x}_1$ and $\boldsymbol{x}_2$. Given a subset $S$ of shared latent factors, it

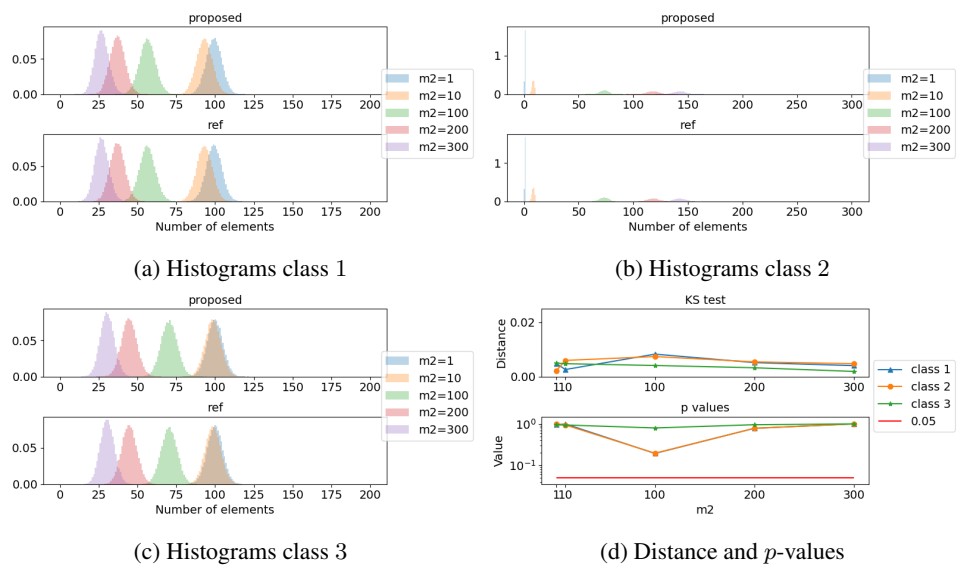

Figure 10: Sensitivity analysis for a varying total number of elements $m_2$ measured with the Kolmogorov-Smirnov test. In this experiment, we have the following specifications: $n = 200$, $\boldsymbol{\omega} = \{1.0, 5.0, 1.0\}$, $m_1 = 200$ and $m_3 = 200$. The values of $m_2$ are given in the plot.

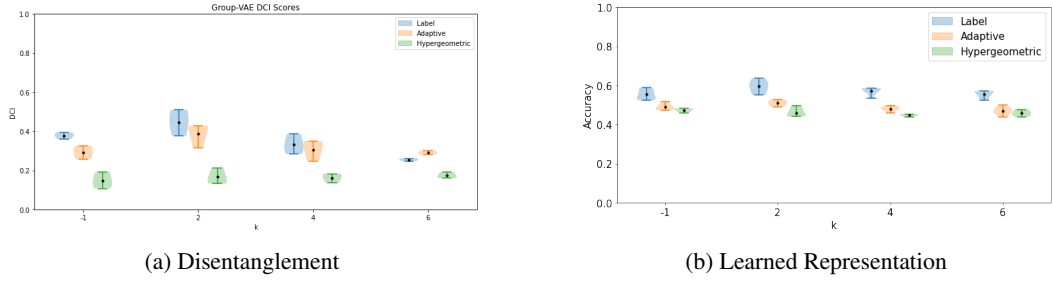

Figure 11: As an additional evaluation, we perform analysis on how disentangled the latent representations are. And related to that, we assess the quality of the learned latent representation using a linear classifier. We see that the dynamics over different $k$ seems to be related for the disentantlement and downstream performance of the learned latent representations.

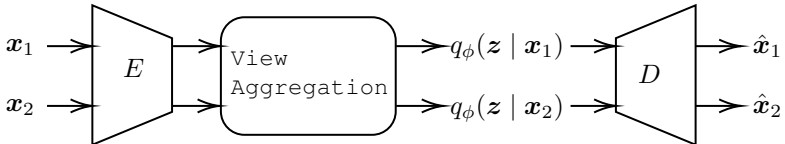

Figure 12: Setup for the weakly-supervised experiment. The three methods differ only in the `View Aggregation` module.

follows

$$q_\phi(z_i \mid \boldsymbol{x}_j) = avg(q_\phi(z_i \mid \boldsymbol{x}_1), q_\phi(z_i \mid \boldsymbol{x}_2)) \qquad \forall \ i \in S \tag{50}$$

$$q_\phi(z_i \mid \boldsymbol{x}_j) = q_\phi(z_i \mid \boldsymbol{x}_j) \qquad \text{else} \tag{51}$$

where $avg$ is the averaging function of choice as described above and $j \in \{1, 2\}$. The methods used (i. e. LabelVAE, AdaVAE, HGVAE) differ in how to select the subset S.

### C.2.2 HYPERGEOMETRICVAE (IN MORE DETAIL)

In our approach (HGVAE), we model the number of shared and independent latent factors of a pair of images as discrete random variables following a hypergeometric distribution with unknown $\boldsymbol{\omega}$. In reference to the urn model, shared and independent factors each correspond to one color and the urn contains $d$ marbles of each color, where $d$ is the dimensionality of the latent space. Given the correct weights $\boldsymbol{\omega}$ and when drawing from the urn $d$ times, the number of each respective color corresponds, in expectation, to the correct number of independent/shared factors. The proposed formulation allows to simultaneously infer such $\boldsymbol{\omega}$ and learn the latent representation in a fully differentiable setting within the weakly-supervised pipeline by Locatello et al. (2019).

To integrate the procedure described above in this framework, we need two additional building blocks. First, we introduce a function that returns $\log \boldsymbol{\omega}$. To achieve this, we use a single dense layer which returns the logits $\log \boldsymbol{\omega}$. The input to this layer is a vector $\boldsymbol{\gamma}$ containing the symmetric version of the KL divergences between pairs of latent distributions, i.e. for latent $P$ and $Q$, the vector contains $\frac{1}{2}(KL(P||Q) + KL(Q||P))$. Second, sampling from the hypergeometric distribution with these weights leads to an estimate $\hat{k}$ and $\hat{s}$. Consequently, we need a method to select $\hat{k}$ factors out of the $d$ that are given. Similar to the original paper, we select the factors achieving the highest symmetric KL-divergence. In order to do this, we sort $\boldsymbol{\gamma}$ in descending order using the stochastic sorting procedure `neuralsort` (Grover et al., 2019). This enables us to select the top $\hat{k}$ independent as well as the bottom $\hat{s} = d - \hat{k}$ shared latent factors. Like AdaVAE, we substitute the shared factors by the mean value of the original latent code before continuing the VAE forward pass in the usual fashion.

### C.2.3 HYPERPARAMETER SENSITIVITY

We perform ablations to examine how sensitive HypergeometricVAE is to certain hyperparameters. We find that the temperature and the learning rate have the most influence on training stability and convergence. First, if we set the temperature $\tau$ too low, we often observe vanishing or exploding gradients. This well-known artifact of the Gumbel-Softmax trick can be avoided by introducing temperature annealing. Hence, similar to the original Gumbel-Softmax trick (Jang et al., 2016; Maddison et al., 2017) and the neuralsort implementation (Grover et al., 2019), we anneal the temperature $\tau$ using an exponential function

$$\tau_t = \tau_{init} \exp(-rt) \tag{52}$$

where $t$ is the current training step, $\tau_{init}$ is the initial temperature, and $r$ is the annealing rate:

$$r = \frac{\log \tau_{init} - \log \tau_{final}}{n_{steps}} \tag{53}$$

$\tau_{final}$ is the final temperature value and $n_{steps}$ is the number of annealing steps. As shown in Figure 13, training loss and shared factor estimation then converge almost independently of the final temperature. In our final experiments, we use identical temperatures $\tau$ for both the differentiable

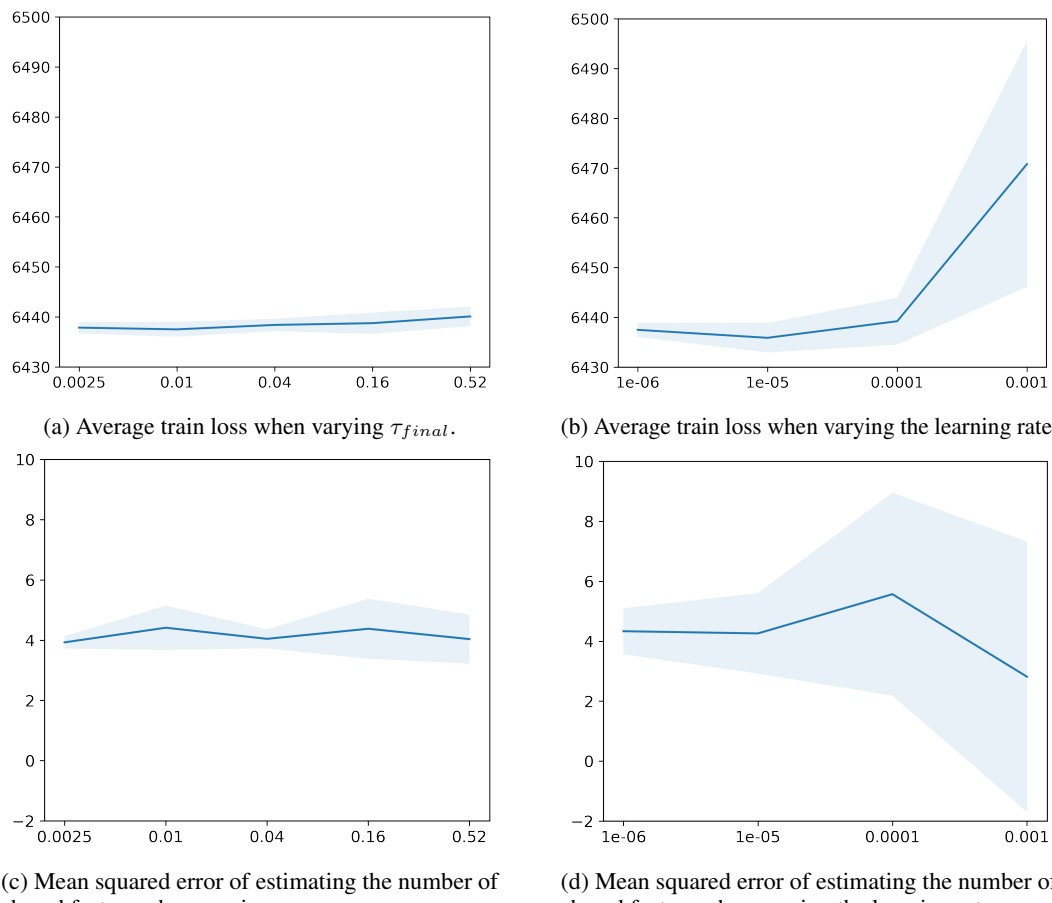

(a) Average train loss when varying $\tau_{final}$.

(b) Average train loss when varying the learning rate.

(c) Mean squared error of estimating the number of shared factors when varying $\tau_{final}$.

(d) Mean squared error of estimating the number of shared factors when varying the learning rate.

Figure 13: Ablation of varying the final temperature $\tau_{final}$ (left) and the learning rate (right) of the HGVAE in the weakly supervised experiment. We made ablations for the training loss (top) and mean squared error of estimating the number of shared factors (bottom). The final temperature had minimal impact on stability and convergence, whereas higher learning rates led to some instabilities.

hypergeometric distribution and neuralsort. We set the initial temperature $\tau_{init}$ to 10 and the final temperature $\tau_{final}$ to 0.01, which is annealed over $n_{steps} = 50000$. Further, we find the learning rate to be the most crucial hyperparameter in terms of convergence. Higher learning rates generally seem to lead to worse training losses. On the other hand, estimating the number of shared factors seems robust on average, although high standard deviations imply decreasing consistency for higher learning rates. We demonstrate this finding in Figure 13. We used an initial learning rate of $10^{-6}$ together with the Adam optimizer (Kingma & Ba, 2014) for our final experiments. Finally, we also experimented with weighting the KL-divergence with a $\beta$ term like in the $\beta$-VAE Higgins et al. (2017), where we did not find an influence on stability and convergence. Hence, we left it at a default value of 1 in our experiments.

### C.2.4 DATA

The mpi3d dataset (Gondal et al., 2019) consists of frames displaying a robot arm and is based on 7 generative factors:

- object color, shape and size
- camera height
- background color
- horizontal and vertical axis

Table 3: We report the runtimes for the three methods used in the weakly-supervised experiment, i. e. labelVAE, adaptiveVAE and hypergeometricVAE. We report the runtime as mean and standard deviation over all runs per experiment. We report the runtimes for the different number of independent factors $k = \{-1, 2, 4, 6\}$.

| METHOD | RUNTIME [S] | | | |
| --- | --- | --- | --- | --- |
| | $k = -1$ | $k = 2$ | $k = 4$ | $k = 6$ |
| LABEL | $21907.4 \pm 273.0$ | $20921.1 \pm 705.9$ | $21389.3 \pm 163.8$ | $21761.1 \pm 567.9$ |
| ADAPTIVE | $29071.5 \pm 133.2$ | $28609.8 \pm 439.9$ | $29479.1 \pm 487.1$ | $29966.3 \pm 303.3$ |
| HYPERGEOMETRIC | $21888.9 \pm 632.9$ | $21299.4 \pm 293.6$ | $21863.9 \pm 190.1$ | $22241.0 \pm 137.8$ |

For more details on the dataset and in general, we refer to `https://github.com/rr-learning/disentanglement_dataset`.

### C.2.5 DOWNSTREAM TASK ON THE LEARNED LATENT REPRESENTATIONS

For the downstream task we sample randomly 10000 samples from the training set and 5000 samples from the test set. For each sample, we extract the predicted shared and the predicted independent parts of both views. Then, for every generative factor of the dataset, three individual classifiers are trained on the respective latent representations of the 10000 training samples. Afterwards, every classifier evaluates its predictive performance on the latent representations of the 5000 test samples. To arrive at the final scores, we extract the prediction of the shared factors on the shared representation and compute the balanced accuracy. Similarly, we calculate the balanced accuracy of the independent factors on the respective independent representation classifiers and average their balanced accuracy. Because the number of classes differs between generative factors we report the adjusted balanced accuracy. We use the implementation from scikit-learn (Pedregosa et al., 2011). For details, see `https://scikit-learn.org/stable/modules/generated/sklearn.metrics.balanced_accuracy_score.html`.

For all shared generative factors, we average the accuracies of the individual classifiers into a single average balanced accuracy. We do the same for the independent factors. This allows us to report the amount of shared and independent information that is present in the learned latent representation. Consequently, we report these averages in the main text.

To evaluate the latent representation we train linear classifiers. More specifically, in this work we use logistic regression classifiers (Cox, 1958) from scikit-learn (Pedregosa et al., 2011). To train the model, we increased the $max\_iter$ parameter so that all models converged and left everything else on default settings.

### C.2.6 RUNTIMES OF DIFFERENT ALGORITHMS

In general, our sampling method scales with the number of classes $\mathcal{O}(c)$ and the sampling for a single class scales with $\mathcal{O}(m_i)$, which is calculated in parallel and a single forward pass. The get a better and empirically validated picture of the overhead created by using the hypergeometric distribution, we also report the training times for the three methods compared. All methods were trained for an equal number of epochs and on identical hardware equipped with NVIDIA GeForce GTX 1080. Table 3 reports the runtimes for all methods and averaged over 5 runs.

We see that the overhead of using the hypergeometric distribution is almost negligible. The proposed hypergeometricVAE reaches approximately equal training runtimes as the labelVAE, which assumes that the number of shared factors is known. It even outperforms the adaptiveVAE, which uses a very simple heuristic, but needs to result to more sophisticated methods in order to avoid cutting gradients in their thresholding function.

## C.3 CLUSTERING

In this section, we provide the interested reader with more details on the clustering experiments. In the following we describe the models, the optimisation procedure and the implementation details used in the clustering task.

### C.3.1 MODEL

We follow a deep variational clustering approach as described by Jiang et al. (2016).

Given a dataset $X = \{\mathbf{x}_i\}_{i=1}^N$ that we wish to cluster into $K$ groups, we consider the following generative assumptions:

$$\mathbf{c} \sim p(\mathbf{c}; \boldsymbol{\pi}), \quad \mathbf{z}_i \sim p(\mathbf{z}_i|c_i) = \mathcal{N}(\mathbf{z}_i|\boldsymbol{\mu}_{c_i}, \boldsymbol{\sigma}_{c_i}^2 \mathbb{I}), \quad \mathbf{x}_i \sim p_\theta(\mathbf{x}_i|\mathbf{z}_i) = Ber(\boldsymbol{\mu}_{x_i}) \quad (54)$$

where $\mathbf{c} = \{c_i\}_{i=1}^N$ are the cluster assignments, $\mathbf{z}_i \in \mathbb{R}^D$ are the latent embeddings of a VAE and $\mathbf{x}_i$ is assumed to be binary for simplicity.

In particular, assuming the generative process described in Equation (54), we can write the joint probability of the data, also known as the likelihood function, as

$$p(X) = \sum_{\mathbf{c}} \int_{\mathbf{z}} p(X, Z, \mathbf{c}) = \sum_{\mathbf{c}} \int_{\mathbf{z}} p(\mathbf{c}; \boldsymbol{\pi}) p(X|Z) p(Z|\mathbf{c}) = \sum_{\mathbf{c}} p(\mathbf{c}; \boldsymbol{\pi}) \prod_i \int_{\mathbf{z}_i} p(\mathbf{x}_i|\mathbf{z}_i) p(\mathbf{z}_i|c_i)$$
$$(55)$$

Different from Jiang et al. (2016), the prior probability $p(\mathbf{c}; \boldsymbol{\pi})$ cannot be factorized as $p(c_i; \boldsymbol{\pi})$ for $i = 1, \ldots, K$ are not independent. By using a variational distribution $q_\phi(Z, \mathbf{c}|X)$, we have the following evidence lower bound

$$\log p(X) \geq E_{q_\phi(Z,c,|X)} \left[ \log \left( \frac{p(\mathbf{c}; \boldsymbol{\pi}) p(X|Z) p(Z|\mathbf{c})}{q_\phi(Z, \mathbf{c}, |X)} \right) \right] = \mathcal{L}_{ELBO}. \quad (56)$$

For sake of simplicity, we assume the following amortized mean-field variational distribution, as in previous work (Jiang et al., 2016; Dilokthanakul et al., 2016):

$$q_\phi(Z, \mathbf{c}|X) = q_\phi(Z|X) q_\phi(\mathbf{c}|X) = \prod_i q_\phi(\mathbf{z}_i|\mathbf{x}_i) q_\phi(c_i|\mathbf{x}_i). \quad (57)$$

From where it follows

$$\mathcal{L}_{ELBO} = E_{q_\phi(Z|X)q_\phi(\mathbf{c}|X)} \left[ \log p(\mathbf{c}|\boldsymbol{\pi}) + \log p(X|Z) + \log p(Z|\mathbf{c}) - \log q_\phi(Z, \mathbf{c}|X) \right] \quad (58)$$
$$= E_{q_\phi(Z|X)q_\phi(\mathbf{c}|X)} \left[ \log p(\mathbf{c}|\boldsymbol{\pi}) \right] + E_{q_\phi(Z|X)} \left[ \log p(X|Z) \right] + E_{q_\phi(Z|X)q_\phi(\mathbf{c}|X)} \left[ \log p(Z|\mathbf{c}) \right]$$
$$- E_{q_\phi(Z|X)} \left[ \log q_\phi(Z|X) \right] - E_{q_\phi(Z|X)q(\mathbf{c}|X)} \left[ \log q_\phi(\mathbf{c}|X) \right]. \quad (59)$$

In the ELBO formulation all terms, except the first one, can be efficiently calculated as in previous work (Jiang et al., 2016). For the remaining term, we rely on the following sampling scheme

$$E_{q_\phi(Z|X)q_\phi(\mathbf{c}|X)} \left[ \log p(\mathbf{c}|\boldsymbol{\pi}) \right] = \sum_{\mathbf{c}} \int_Z q_\phi(Z|X) q_\phi(\mathbf{c}|X) \log p(\mathbf{c}|\boldsymbol{\pi})) \quad (60)$$

$$= \sum_{\mathbf{c}} \prod_i \int_{\mathbf{z}_i} q_\phi(\mathbf{z}_i|\mathbf{x}_i) q_\phi(c_i|\mathbf{x}_i) \log p(\mathbf{c}|\boldsymbol{\pi}) \quad (61)$$

$$= \sum_{l=1}^L \log p(\mathbf{c}^l|\boldsymbol{\pi}), \quad (62)$$

where we use the SGVB estimator and the Gumbel-Softmax trick (Jang et al., 2016) to sample from the variational distributions $q_\phi(\mathbf{z}_i|\mathbf{x}_i)$ and $q_\phi(c_i|\mathbf{x}_i)$ respectively. The latter is set to a categorical distributions with weights given by:

$$p(c_i|\mathbf{z}_i; \boldsymbol{\pi}) = \frac{\mathcal{N}(\mathbf{z}_i|\boldsymbol{\mu}_{c_i}, \boldsymbol{\sigma}_{c_i}^2) \pi_{c_i}}{\sum_k \mathcal{N}(\mathbf{z}_i|\boldsymbol{\mu}_k. \boldsymbol{\sigma}_k^2) \pi_k}, \quad (63)$$

$L$ is the number of Monte Carlo samples and it is set to 1 in all experiments.

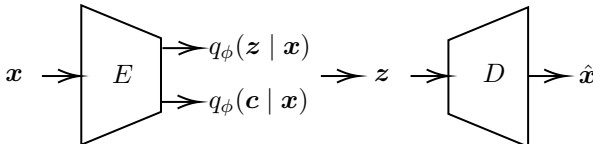

Figure 14: General Architecture for the clustering experiments. All methods have the same architecture details. They only differ in their definition of the prior distribution $p(c; \pi)$.

### C.3.2 IMPLEMENTATION DETAILS

To implement our model we adopted a feed-forward architecture for both the encoder and decoder of the VAE with four layers of $500, 500, 2000, D$ units respectively, where $D = 10$. The VAE is pretrained using the same layer-wise pretraining procedure used by Jiang et al. (2016). Each data set is divided into training and test sets, and all the reported results are computed on the latter. We employed the same hyper-parameters for all experiments. In particular, the learning rate is set to $0.001$, the batch size is set to $128$ and the models are trained for $1000$ epochs. Additionally, we used an annealing schedule for the temperature of the Gumbel-Softmax trick. As the VaDE is rather sensitive to initialization, we used the same pretraining weights provided by Jiang et al. (2016). These weights have been selected by the baseline to enhance the performance of their method, leading to an optimistic outcome. If a random initialization were used instead, the clustering performance would be lower. Nonetheless, the focus of our work is on comparing methods rather than on absolute performance values. Figure 14 displays the general architecture of all methods. The methods only differ in their definition of the prior probability distribution $p(c; \pi)$.

