# OpenReview forum: "Learning Group Importance using the Differentiable Hypergeometric Distribution"
_ICLR.cc/2023/Conference — ICLR 2023 notable top 25%_

### Official Review · Reviewer_xbJ8 · 2022-10-22

**Confidence:** 3
**Correctness:** 4
**Technical Novelty And Significance:** 3
**Empirical Novelty And Significance:** 3
**Recommendation:** 8

**Clarity, Quality, Novelty And Reproducibility:**

The manuscript is well written and clear.
While the approach is constructed of well known pieces; conditional sampling and the Gumbel-Softmax trick, they are used in a smooth and elegant fashion. As a consequence, I find the work novel and potentially impactful.


**Strength And Weaknesses:**

Modeling categorical data using neural architectures has always been a challenge. The proposed method extends the tools available to model different classes of problems. Additionally, this is accomplished without significantly increasing the number of parameters.

My only concern with this work is the issue of fitting the importance weights. The authors do not seem to spend any time discussing the implications of this method on the convergence of gradient optimization routines. Can this method cause exploding or vanishing gradients? If so, under what conditions? Some remarks, analysis, or empirical evidence on this topic would be welcome.


**Summary Of The Paper:**

The authors propose a differentiable formulation of the hypergeometric distribution using the Gumbel-Softmax trick. They provide a sampling algorithm and demonstrate its use on two tasks that require modeling of discrete distributions. The first is a weakly-supervised representation learning problem for video frames. The second is a deep variational clustering application. In both applications, the use of the differentiable hypergeometric distribution improves performance. Additionally, the authors demonstrate that for a wide range of parameter values, one cannot distinguish between the a baseline non-differentiable implementation of the hypergeometric distribution and the proposed differentiable implementation using a KS test.


**Summary Of The Review:**

The authors propose a useful tool for modeling discrete problems in differentiable frameworks.

---

### Official Review · Reviewer_yMNr · 2022-10-24

**Confidence:** 2
**Correctness:** 4
**Technical Novelty And Significance:** 3
**Empirical Novelty And Significance:** 2
**Recommendation:** 6

**Clarity, Quality, Novelty And Reproducibility:**

Clarity, Quality, Novelty, Reproducibility

* Clarity could be improved in the applications.
* The rest seem fine, although my read on novelty in this area is limited.


**Strength And Weaknesses:**

Strengths

* Seems like a potentially useful tool.


Weaknesses

* Methodology could still use further investigation. The Kolmogorov-Smirnov test is one nice step, but it would also be nice to mathematically quantify the difference between the hypergeometric and the proposed distribution with standard measures of distributional similarity (various divergences, TV distance, etc.).


Questions

* It seems like this methodology is most applicable for variational inference, which combines gradient optimization and distributional inference. Are there other methodological applications I should have in mind?

**Summary Of The Paper:**

This paper proposes an approximation of the hypergeometric distribution that introduces a differentiable hyperparameter. The benefit is that, in machine learning algorithms where data may fall into discrete categories, the number and size of the categories is potentially learnable with gradient methods. The paper illustrates this distribution numerically in two settings: learning the number of dependent components in a weak supervision setting and clustering.

**Summary Of The Review:**

Summary

* Potentially a useful methodology for working with distributions having underlying categories in some situations.

---

### Official Review · Reviewer_uGmy · 2022-10-24

**Confidence:** 3
**Correctness:** 4
**Technical Novelty And Significance:** 3
**Empirical Novelty And Significance:** 4
**Recommendation:** 8

**Clarity, Quality, Novelty And Reproducibility:**

The paper is so well written and provides a comprehensive treatment on the matter with sufficient demonstrations of the practical value that I do not have any remarks concerning the content. The technical content is not terribly complicated and one could argue that this is a fairly natural continuation of the earlier relaxations, but nevertheless something that has not been done yet and the devil is often in the details that probably were not clear before all the effort that went to developing this. I commend the authors on being very clear on the limitations and even small technical details that will be important for practical implementations (e.g. clearly saying how log-domain makes calculations more stable etc).

**Strength And Weaknesses:**

Strengths:
- Very clear contribution in form of a practical tool that can be used by others
- Potential for high impact, in a similar manner as the concrete/GS distribution had an impact on how categorical variables are used in broad range of models
- Very well written for a paper this technical

Weaknesses:
- Nothing really, but a graphical illustration of the hypergeometric distribution and the role of the augmentation would not hurt; it would be especially useful if this becomes a core reference for people not so familiar with the literature

**Summary Of The Paper:**

The paper proposes a differentiable approximation of the hypergeometric distribution and shows that it works well.

**Summary Of The Review:**

Well written comprehensive treatment of a practical tool that makes building models that rely on the hypergeometric distribution easier.

---

### Official Review · Reviewer_gdaG · 2022-10-25

**Confidence:** 2
**Correctness:** 3
**Technical Novelty And Significance:** 3
**Empirical Novelty And Significance:** 4
**Recommendation:** 8

**Clarity, Quality, Novelty And Reproducibility:**

For the most part the paper is very readable and most individual points are reasonably clear, however, as hinted above the paper could benefit from an extremely simple application in which the optimisation which is rendered possible by the proposal is described explicitly.  Having said this, the actual description of the algorithm, and the sampling procedure for generating realisations from the distribution is clear and, having some experience with other machinery described in the experiments I believe the results are reproducible.

**Strength And Weaknesses:**

Strengths:
- The work investigates a relatively under-studied topic in machine learning where alternatives perform modelling based on i.i.d. sampling and may, at best, use post-hoc assessment of group sizes or similar to infer importance weights. By building the importance weights into the estimation both inference and estimation can be improved.
- The empirical results show that the method can offer improvements on existing methods in some interesting and relevant problems.

Weaknesses:
- The main weakness of the paper is that it expects a lot of knowledge from the reader, and many "for details see..." pieces of text which may make the work not stand well along as a single item of literature.
- There are also potential issues with clarity, where while the individual points are clear it is not immediately forthcoming how the optimisation rendered possible by the differentiability of the mass function is implemented.

**Summary Of The Paper:**

The paper proposes an approximation of the multivariate non-central hypergeometric distribution in which, first the multivariate distribution is expressed through the product rule as a product of conditional distributions, each of which can be treated as a univariate non-central hypergeometric distribution by grouping the remaining (un-accommodated/sampled) classes into one. Finally, these univariate non-central hypergreometric distributions are approximated using the Gumbel-Softmax approximation technique. This makes the differentiation of the joint distribution with respect to the class importance weights feasible.

The paper concludes with some experiments; first showing the accuracy of the approximation through simulating from the true and approximate distributions, and then in two real applications where it shows superiority to some existing methods in learning numbers of shared and independent latent generative factors from coupled image observations and in clustering, where the proposed distribution is used as a tunable prior for a VAE.

**Summary Of The Review:**

The paper covers an interesting and under-explored topic in machine learning, and the proposed method seems to offer options for modelling, estimation and inference which are otherwise quite limited to heuristics. Although the paper is very readable and the details are well explained, as a reader one might feel still somewhat at a loss for some of the higher level practical aspects of the proposed approximation.
 The experiments show that the proposed distribution has realisable benefits in some important and interesting applications over existing alternatives and more naive straightforward formulations.

---

### Decision · Program_Chairs · 2023-01-20

**Decision:**

Accept: notable-top-25%

**Justification For Why Not Higher Score:**

see meta-review

**Justification For Why Not Lower Score:**

see meta-review

**Metareview: Summary, Strengths And Weaknesses:**

This work proposes a novel method for differentiable hypergeometric distributions over subsets.

The reviewers and I agree that this is a very good submission that presents new, interesting ideas, in a well-exposed manner.

**Note From Pc:**

if the above contains the word "oral" or "spotlight" please see: "oral" presentation means -> notable-top-5% and "spotlight" means -> notable-top-25%. As stated in our emails, we are disassociating presentation type from AC recommendations